# Version 8 IMK/IAA MIPAS temperatures from 12–15 $\mu$m spectra: Middle and Upper Atmosphere modes

Maya García-Comas[1,*], Bernd Funke[1,*], Manuel López-Puertas[1], Norbert Glatthor[2], Udo Grabowski[2], Sylvia Kellmann[2], Michael Kiefer[2], Andrea Linden[2], Belén Martínez-Mondéjar[1], Gabriele P. Stiller[2], and Thomas von Clarmann[2]

[1]Instituto de Astrofísica de Andalucía, CSIC, Granada, Spain
[2]Karlsruhe Institute of Technology, Institute of Meteorology and Climate Research, Karlsruhe, Germany

**Correspondence:** Maya García-Comas (maya@iaa.es)

**Abstract.** Motivated by an improved ESA version of MIPAS calibrated spectra (version 8.03), we have released version 8 of MIPAS temperatures and pointing information retrieved from 2005–2012 MIPAS measurements at 12–15 $\mu$m in the Middle Atmosphere (MA), Upper Atmosphere (UA) and Noctilucent Cloud (NLC) measurement modes. The IMK/IAA retrieval processor in use considers non-local thermodynamic equilibrium (non-LTE) emission explicitly for each limb scan. This non-LTE treatment is essential to obtain accurate temperatures above the mid-mesosphere, because at the altitudes covered, up to 115 km, the simplified climatology-based non-LTE treatment employed for the nominal (NOM) measurements is insufficient. Other updates in MA/UA/NLC V8 non-LTE temperature retrievals from previous data releases include: more realistic atomic oxygen and carbon dioxide abundances; an updated set of spectroscopic data; an improved spectral shift retrieval; a continuum retrieval extended to altitudes up to 58 km; consideration of an altitude-dependent radiance offset retrieval; the use of wider microwindows above 85 km to capture the offset; an improved accuracy in forward model calculations; new temperature a priori information; improved temperature horizontal gradient retrievals; and, the use of MIPAS version 5 interfering species, where available. The resulting MIPAS MA/UA/NLC IMK/IAA temperature dataset is reliable for scientific analysis in the full measurement vertical range for the MA (18–102 km) and the NLC (39–102 km) observations, and from 42 to 115 km for the UA observations. The random temperature errors, dominated by the instrumental noise, are typically less than 1 K below 60 km, 1–3 K at 60–70 km, 3–5 K at 70–90 km, 6–8 K at 90–100 km, 8–12 K at 100–105 km and 12–20 K at 105–115 km. Pointing correction random errors, also mainly arising from instrumental noise, are on average 50 m for tangent altitudes up to 60 km and decrease linearly to values smaller than 20 m for altitudes above 95 km. The vertical resolution is 3 km at altitudes below 50 km, 3–5 km at 50–70 km, 4–6 km at 70–90 km, 6–10 km at 90–100 km and 8–11 km at 100–115 km. The systematic errors of retrieved temperatures below 75 km are driven by uncertainties in the $CO_2$ spectroscopic data and, above 80 km, by uncertainties in the non-LTE model parameters (including collisional rates and atomic oxygen abundance) and the $CO_2$ abundance. These lead to systematic temperature errors of less than 0.7 K below 55 km, 1 K at 60–80 km, 1–2 K at 80–90 km, 3 K at 95 km, 6–8 K at 100 km, 10–20 K at 105 km and 20–30 K at 115 km. Systematic errors in the tangent altitude correction, mainly arising from $CO_2$ spectroscopic uncertainties, are 250 m at 20 km and 200 m at 40–60 km, 100 m at 80 km and smaller than 50 m above 90 km. The consistency between the MA/UA/NLC and the NOM IMK/IAA datasets is excellent below 70 km (typ-

ical 0.5–1 K differences). The comparison of this V8 temperature dataset with co-located SABER temperature measurements shows an excellent agreement, even better than in previous MIPAS IMK/IAA versions.

## 1 Introduction

The European Space Agency (ESA) has released version 8.03 of calibrated spectra from the Michelson Interferometer for Passive Atmospheric Sounding (MIPAS), that flew onboard the ENVISAT satellite from February 2002 to April 2012. One of the main upgrades of the L1b version 8.03 with respect to previous 5.02/5.06 spectra is the use of a time-dependent correction of the non-linearity of the detector response for the calibration instead of relying on the preflight characterization (Birk and Wagner, 2010; Kleinert et al., 2018).

MIPAS recorded the Earth's global limb emission from 4.1 to 14.7 $\mu$m (685–2410 $cm^{-1}$) under full spectral resolution (FR; 0.025 $cm^{-1}$, unapodized) since 2002 to March 2004, and under reduced spectral resolution (RR; 0.0625 $cm^{-1}$, unapodized) since then to April 2012 (Fischer et al., 2008). MIPAS vertical coverage went approximately from 6 to 70 km in its NOMinal (NOM) mode of observation. It was extended up to the lower thermosphere in three special observation modes: up to 102 km in its Middle Atmosphere (MA) and its its NoctiLucent-Cloud (NLC) modes, and up to 170 km in Upper Atmosphere mode (UA). The MA and UA measurements were taken during approximately two days every ten days, and the NLC measurements were taken a few days in a row every year during the noctilucent cloud seasons (polar summers in both hemispheres). The vertical sampling in the mesosphere for these three special modes of observation was better than in NOM (every 3 km in the former vs. 4 km in NOM) and the NLC mode sampling was even better around the mesopause (every 1.5 km).

Apart from the operational MIPAS retrievals from ESA based on the Level 2 processor Optimised Retrieval Model (ORM) (see e.g., Raspollini et al., 2013), atmospheric temperature has also been retrieved in the past from previous L1b versions of MIPAS $CO_2$ spectral lines around 12–15 $\mu$m using the IMK/IAA processor (NOM mode: von Clarmann et al. (2009b); Stiller et al. (2012)); MA, UA and NLC modes: García-Comas et al. (2012); García-Comas et al. (2014)). The main differences of the IMK/IAA temperature retrievals with respect to those of ESA were the selected spectral micro-windows; the regularization approach; the consideration of temperature horizontal gradients along the line of sight instead of the approximation of local horizontal homogeneity, the cloud filtering threshold, and the simultaneous retrieval of a pointing correction instead of correcting the pointing using the simultaneously retrieved pressures. In addition, the temperature retrieval of the MA, UA and NLC mode measurements with the IMK-IAA processor considered the $CO_2$ emissions in non-Local Thermodynamic Equilibrium (non-LTE). The latter is crucial in the Mesosphere and Lower Thermosphere (MLT) region, particularly, in the high latitude summer (López-Puertas and Taylor, 2001).

The release of version 8 spectra provided a clear motivation to reprocess the data and take the opportunity to improve the retrievals in several ways. Regarding the ESA operational temperature retrievals, these improvements are described in Dinelli et al. (2021) and Raspollini et al. (2022), and include, among other developments, the consideration of temperature horizontal gradients taken from the ECMWF ERA-Interim data. ESA temperature retrievals are performed for all observation modes.

Raspollini et al. (2021) reported that the upper boundary of the scientifically useful vertical range of their temperature profiles retrieved from observations in the MA, UA and NLC modes is 78 km.

Regarding the temperature retrievals with the IMK/IAA processor, Kiefer et al. (2021) describe the recently retrieved temperatures based on MIPAS 8.03 spectra taken in the NOM mode and the improvements with respect to previous versions. Among other changes, the IMK/IAA NOM retrievals now consider non-LTE. Instead of full non-LTE modeling for each limb scan, NOM retrievals (extending just to the lower mesosphere) only demand the use of seasonal and latitudinal climatologies of temperature-parameterized $CO_2$ vibrational level populations.

However, MIPAS temperature retrievals that extend to the lower thermosphere require explicit non-LTE calculations tailored to the specific atmospheric conditions. Therefore, for MA, UA and NLC temperature and pointing retrievals, we use the IMK/IAA processor incorporating the retrieval approach described in Kiefer et al. (2021) supplemented with full non-LTE modeling. That is to say, we calculate the $CO_2$ vibrational level populations individually for each limb scan on the basis of the actual atmospheric conditions instead of using climatological data. With this scheme, we have re-processed MIPAS Level 1b version 8.03 MA, UA and NLC RR spectra, spanning from January 2005 to April 2012, to provide the V8R_T_561, V8R_T_661 and V8R_T_761 temperatures and line of sight information dataset (see Sect. 2). Although the inclusion of full non-LTE slows down the data processing as compared with the non-LTE parameterization used in NOM retrievals, it is essential for obtaining accurate temperatures above the mid-mesosphere. We note that a small percentage of the Level 1b data version used has problems around 85 km, which prevented Level 2 processing. In Sect. 2, we discuss the improvements of the new database with respect to preceding versions achieved by the new retrieval settings. We also provide a summary of the temperature systematic and random errors (Sect. 3), that were estimated for 34 atmospheric scenarios (see Supplement). Further, we compare the results with IMK/IAA V5 MA/UA/NLC temperatures derived using version 5.02/5.06 MIPAS spectra (Sect. 4) and IMK/IAA V8 NOM temperatures derived using version 8.03 spectra (Sect. 5). We finally show comparisons of this new temperature dataset with measurements from SABER.

## 2 V8 MA/UA/NLC temperature retrievals

MIPAS vertical coverage extended up to the lower thermosphere in three special observation modes. In its Middle Atmosphere mode (MA), it measured the limb approximately from 18 to 102 km in 3 km increments. In its Upper Atmosphere mode (UA), it covered tangent altitudes from 42 to 170 km in 3 km steps below 102 km and 5 km steps above. In its NoctiLucent-Cloud mode (NLC), the tangent heights of the sweeps ranged approximately from 39 to 102 km in 3 km steps, except from 78 to 87 km where MIPAS tangent altitudes were spaced 1.5 km. MIPAS MA/UA/NLC measurements were taken systematically since January 2005 to April 2012. At that time, MIPAS operated in its reduced spectral resolution, i.e., at 0.0625 $cm^{-1}$ (unapodized; Fischer et al., 2008).

Previously, we have distributed two sets of versions of temperatures retrieved from MIPAS measurements in the MA/UA/NLC modes using the IMK/IAA processor and considering full non-LTE. The first set consisted in versions V4O_T_511 (MA), V4O_T_611 (UA) and V4O_T_711 (NLC), or just version V4O_T_m11 referring to them jointly (García-Comas et al., 2012).

They were based on ESA version 4.65/4.67 spectra. The second set of temperatures consisted in versions V5R_T_521 (MA), V5R_T_621 (UA) and V5R_T_721 (NLC), or just version V5R_T_m21 altogether (García-Comas et al., 2014), which were based on ESA version 5.02/5.06 spectra. The release of the improved version 8.03 spectra by ESA highlighted the necessity for a re-processing of the data, leading to the development of the new IMK-IAA version 8 temperature retrievals presented here. This consists of versions V8R_T_561 (MA), V5R_T_661 (UA) and V5R_T_761 (NLC), or just collectively referred to as version V8R_T_m61. Note that the reference to V8R_T_m61 temperatures in the remainder of this paper intentionally excludes temperatures derived from NOM mode measurements, due to the additional requirements for retrievals above the mid-mesosphere.

Except for the treatment of non-LTE and regularization above 70 km, the IMK-IAA V8 retrieval configuration used to derive temperatures and pointing information from MIPAS MA, UA and NLC mode 12–15$\mu$m spectra (V8R_T_m61) is the same as that used for NOM measurements (V8R_T_260) and described in Kiefer et al. (2021). In short, the frequency shift is first derived from the spectra prior to the temperature retrieval. Then, temperatures are jointly retrieved with a correction to the ESA's engineering tangent heights of the line of sight from selected spectral microwindows using regularization. For temperature, we apply a constrained nonlinear least squares fitting using a first-order difference Tikhonov regularization, making use of an additional weak diagonal constraint around the mesopause for the MA/UA/NLC modes. For the tangent height correction, we use optimal estimation with a very strong constraint at 105 km to meet ESA's engineering tangent heights. A hydrostatical reconstruction is performed in each iteration.

Additionally, MA, UA and NLC temperatures are retrieved considering a sophisticated $CO_2$ non-LTE scheme as described in Funke et al. (2012) (unlike NOM retrievals, which use a parameterized non-LTE climatology). The algorithm jointly fits interfering species (namely, ozone and water vapor) and also retrieves the horizontal temperature gradients, the background continuum up to 58 km, and an altitude-dependent radiance offset profile. A list of the main aspects of the retrieval baseline follows. Where relevant, we also mention the updates with respect to version V5R_T_m21 retrievals, the previous release of MIPAS IMK/IAA MA, UA and NLC temperatures (García-Comas et al., 2014). For further details on the V8 retrieval set-up, we refer the reader to Sect. 3 of Kiefer et al. (2021).

−*MIPAS spectra*: The version of the calibrated MIPAS spectra used is the latest supplied by ESA (8.03), which uses a time-dependent correction of the non-linear detector response for the calibration (Birk and Wagner, 2010; Kleinert et al., 2018). Version 5.02/5.06 was used for preceeding IMK/IAA V5 retrievals, which used a preflight characterization. Also, gain calibration has been taken for the corresponding day in V8, whereas it was taken once per week in previous versions.

−*Microwindows and spectroscopic data*: Table 1 shows the spectral intervals, referred to as microwindows, from where temperature and altitude information of the line of sight is retrieved. This selection of microwindows is almost identical to that used in previous versions of MA, UA and NLC temperature retrievals (García-Comas et al., 2014), except that the width of some microwindows comprising fundamental band lines has been slightly extended at high altitudes to better capture the continuum and the radiance offset. In the seek for consistency with MA, UA and NLC temperatures, the V8 NOM temperature retrievals also adopted these microwindows, which was not the case in previous versions. The spectroscopic database used for most

**Table 1.** Microwindows used in V8R_T_m61 MIPAS MA, UA and NLC mode temperature retrievals.

| Wavenumber ($cm^{-1}$) | Altitude (km) |
|---|---|
| 686.8125−689.7500 | 42−120 |
| 689.8750−692.6250 | 42−120 |
| 699.4375−702.3750 | 42−120 |
| 719.6250−722.5000 | 33−120 |
| 731.2500−731.8125 | 21−72 |
| 740.3750−742.8750 | 33−69 |
| 744.3125−745.5000 | 21−72 |
| 748.9375−749.8125 | 20−72 |
| 765.8750−766.5625 | 21−72 |
| 780.4375−780.6250 | 20−73 |
| 791.1875−792.6875 | 20−63 |
| 798.1250−798.5000 | 21−72 |
| 810.8125−811.0625 | 20−72 |
| 812.2500−812.5625 | 20−72 |

species in V8R_T_m61 is HITRAN 2016 (Gordon et al., 2017). We used Hitran 2008 in V5R_T_m21. For $O_3$ and $HNO_3$, we use a dedicated MIPAS spectroscopic database (Flaud et al., 2003).

−*Frequency shift determination*: Even though the MIPAS Level 1b data are spectrally calibrated, our retrieval requires adjustments to account for shifts due to the modeling of the instrument line shape. To solve this issue, we derive a mean frequency shift scale from MIPAS spectra at 38 km of a full orbit each day prior to the temperature retrieval. To avoid noise fluctuations, the spectral shift is retrieved in two iterations. In the first step, the spectral shift is retrieved in a maximum likelihood approach from each individual measurement. The second step uses a maximum a posteriori scheme constrained to the temporal mean and variances of the results from the first iteration. The microwindows used for the frequency shift retrieval are listed in Table 1 of Kiefer et al. (2021). We had to include $NO_2$ in the forward model because of a significant mesospheric contribution.

−*Continuum*: Following the findings of Haenel et al. (2015), we now jointly fit the continuum up to 58 km, instead of up to 33 km in the V5 retrievals, which eliminates biases and improves convergence. We use a zero continuum a priori.

−*Offset*: We derive an altitude-dependent radiance offset for each microwindow to correct for the zero radiance level, constrained to the empirical profile derived by Kleinert et al. (2018). The offset correction is indistinguishable from the continuum above 60 km, leading to problems with their simultaneous retrieval. We solve this problem by strongly constraining the offset to its a priori profile.

**Table 2.** Values prescribing the dependence with altitude of the smoothing regularization terms used in version 8 MA, UA and NLC mode temperature retrievals.

| Altitude (km) | $\gamma_S$ ($K^{-2}$) |
|---|---|
| 0−40 | 0.7 |
| 45 | 0.72 |
| 50 | 0.7 |
| 60 | 0.5 |
| 70 | 0.2 |
| 80-120 | 0.1 |

−*Temperature regularization*: The IMK/IAA MA/UA/NLC temperatures are retrieved from the ground to 120 km at 1 km steps up to 50 km, 2 km steps from 50 to 100 km, 2.5 km steps from 100 to 105 km and 5 km steps from 105 to 120 km. Both NOM and MA/UA/NLC retrievals include a first-order difference Tikhonov regularization that constrains the vertical temperature gradients. Note that, unlike optimal estimation or maximum a posteriori retrievals (Rodgers, 2000), the regularization scheme chosen here does not push the temperatures towards the a priori profiles but only constrains the shape of the temperature profiles. MA/UA/NLC retrievals further use a weak diagonal temperature constraint around 90 km in just to prevent unreliable values in the cold polar summer mesopause. Table 2 summarizes the so-called $\gamma$ values (see Eq. 2 of Kiefer et al. (2021)), which govern the altitude variation of the regularization in the MA/UA/NLC retrievals.

−*Temperature, pressure and tangent altitude a priori information*: As in V8R_T_260 NOM retrievals, temperature a priori information in V8R_T_m61 MA, UA and NLC retrievals is taken from ECMWF ERA Interim reanalysis fields (Dee et al., 2011) smoothly merged between 43 and 53 km towards bias-corrected Specified-Dynamics Whole Atmosphere Community Climate Model version 4 (SD-WACCM4) (Garcia et al., 2017) at MIPAS geolocations and times. We performed the WACCM temperature bias correction by using the V5R_T_m21 2005-2012 temperature composites (García-Comas et al., 2014). A priori tangent heights come from the line-of-sight engineering information. Pressure is hydrostatically reconstructed above the lowest tangent height, starting with the ECMWF value at that altitude. The temperature a priori was taken from ECMWF below 65 km and NRLMSISE-00 (Picone et al., 2002) above 65 km in V5R_T_m21.

−*Tangent altitude regularization*: In contrast to the temperature regularization, for the tangent altitudes a maximum-a-priori-type regularization (Rodgers, 2000) is used, which pushes the retrieved tangent altitudes towards the a priori information. The tangent altitudes are constrained towards ESA's line-of- sight engineering information assuming a 60 m standard deviation in the relative pointing between adjacent tangent altitudes, and a 900 m absolute pointing uncertainty, allowing for a vertical shift of the entire limb scan. In the MA/UA/NLC retrievals there is an additional very hard constraint at 105 km towards the engineering values. This update with respect to version 5 retrievals is motivated by the results of Jurado-Navarro et al. (2016).

They retrieved very small deviations of the tangent altitudes in the lower thermosphere from the engineering tangent altitudes using MIPAS 4.3 $\mu$m measurements.

$-$*Trace gas distributions*: The $CO_2$ abundances used to simulate the spectra are extracted from SD-WACCM4 climatology, while $H_2O$ and $O_3$ abundance profiles are taken from MIPAS version 5 retrievals (García-Comas et al., 2016; López-Puertas et al., 2018). Regarding atomic oxygen, we have revised the calculation scheme in order to use more realistic abundances than those in the V5 retrievals, where O came from NRLMSISE-00 (note that atomic oxygen mainly affects the non-LTE $CO_2$ populations). The $O_x$ and H used to derive O are taken from bias-corrected WACCM4 at MIPAS geolocations. The bias correction for $O_x$ is estimated using the MIPAS $O_3$ daytime climatology (versions V5_O3_522; López-Puertas et al. (2018)) under consideration of photochemical equilibrium.

In addition, since the WACCM bias is estimated from daytime measurements, a tidal correction is applied to account for day/night differences as described in Sect. 2.2.3 of López-Puertas et al. (2023). The bias correction for H, required for nighttime measurements, is obtained from the ratio of MIPAS V5 day- and nighttime $O_3$ climatologies as also described in Sect. 2.2.3 of López-Puertas et al. (2023).

Finally, O is calculated from the bias-corrected $O_x$ and H by means of photochemical box modeling using ozone photoabsorption coefficients based on the Tropospheric Ultraviolet and Visible (TUV) radiation model version 5.2 driven by the Climate Model Intercomparison Project round 6 (CMIP6) solar irradiance variations. This approach essentially results in a climatological atomic oxygen as derived from MIPAS with transient variability as provided by WACCM. The atomic oxygen derived in this way is reliable below 97 km. Above 97 km, we use the atomic oxygen from SD-WACCM4 simulations at MIPAS geolocations.

$-$*Temperature horizontal inhomogeneities retrieval*: von Clarmann et al. (2009a) showed that MIPAS limb radiances originate up to $\pm$400 km around the tangent point. We therefore derive horizontal temperature gradients as described in Kiefer et al. (2021). We prescribe a full 3D temperature field constructed from ECMWF ERA-Interim temperatures below 60 km and NRLMSISE-00 temperatures above and we retrieve the linear corrections to this field. V5R_T_m21 retrievals only allowed for a linear correction of horizontal temperature gradients from ECMWF-40.

$-$*Forward model numerical settings*: The spectral grid for the monochromatic radiance calculations is $5 \times 10^{-4}$ cm$^{-1}$. The computational accuracy of the absorption coefficient calculations in the forward model is now improved with respect to V5R_T_m21. We use five pencil beams to numerically integrate the signal in the vertical direction over the field of view.

$-$*Non-LTE treatment*: The most salient difference between the V8R_T_261 NOM mode temperature retrievals described in Kiefer et al. (2021) and the V8R_T_m61 MA, UA and NLC mode retrievals described here is that, instead of using climatological $CO_2$ vibrational level populations, we explicitly calculate the $CO_2$ vibrational level populations for the atmospheric conditions of each scan. This is essential for the accurate retrieval of temperatures above 70 km. The non-LTE population calculations are performed in each iteration of the retrieval loop with the GRANADA algorithm and the non-LTE setup described in Funke et al. (2012). The non-LTE collisional rates affecting 4.3 $\mu$m $CO_2$ levels change slightly with respect to those in Funke et al. (2012) using the values derived from Jurado-Navarro (2015) but this has a negligible impact on the 15 $\mu$m levels. We

also correct the non-LTE populations along the line of sight according to the simultaneously retrieved horizontal temperature gradient, which is particularly relevant when the line of sight crosses the poles during the solstices. The line-by-line Karlsruhe Optimized and Precise Radiative transfer Algorithm (KOPRA) forward model (Stiller et al., 2002), one of the processor modules which is internally interfaced with the Generic RAdiative traNsfer AnD non-LTE population Algorithm (GRANADA), then simulates the 15$\mu$m $CO_2$ emission using these non-LTE populations. The non-LTE scheme and collisional rates in these V8 retrievals is the same as in the previous V5 retrievals.

Figure 1 shows the 2007-2012 composite of V8 MA/UA zonal mean temperatures for each calendar month. They correspond to the zonal means of all available MIPAS measurements at geolocations within ±5° in latitude to each grid point. We do not include in this figure NLC and 2005-2006 MA/UA temperatures because they do not provide a regular temporal coverage along the natural year. We will focus here more on the MIPAS temperatures in the MLT region, since the stratosphere was well covered by the NOM measurements, and Kiefer et al. (2021) have already discussed the stratospheric temperatures. We present temperatures up to 115 km. Several features related to well-known mechanisms are evident in Fig. 1:

- Lower-thermospheric temperatures are highest during the polar summer (300 K at 105 km and 370 K at 115 km) and lowest during the polar winter and the equatorial equinoxes (210 K at 105 km and 330 K at 115 km) due to the corresponding variation of total insolation.

- The meridional circulation, controlled by gravity wave filtering, causes the positive summer-to-winter latitudinal gradients of mesopause temperature and mesopause altitude (Garcia and Solomon, 1985) and leads to MIPAS mesopauses at 130–140 K and 86–88 km in polar summer, and at 180 K and 98 km in polar winter.

- The mesospheric semi-annual oscillation (SAO), caused by the filtering of gravity waves by the stratospheric wind SAO (Dunkerton, 1982), is responsible for the local maximum at 85 km at low latitudes (at 80 km in March) that occurs during the equinox months and leads to peak temperatures of 220 K in April.

- The typical polar summer transition from maximum ozone heating in the low mesosphere to maximum cooling from upwelling in the high mesosphere results in the largest mesospheric temperature vertical gradients, which maximize in June and July in the Northern Hemisphere (NH), and in December and January in the Southern Hemisphere (SH).

- The average mesospheric vertical gradients of heating due to downwelling during the polar night winter months result in minimum mesospheric temperature vertical gradients in February in the NH and May-June in the SH. Note that this average behavior in the NH is influenced by the occurrence of elevated stratopause events, where the temperature vertical gradients become positive.

- The vertical deflection of the polar winter vortex to lower latitudes (Fleming et al., 1990) and the associated diabatic descent from gravity wave-driven circulation (Hitchman et al., 1989) result in the wrinkled mid-to-upper mesospheric temperature contours from May to August in the SH and from October to March in the NH. These even lead to a distinct warmer mesopause centered around 50°N–60°N in December and January in MIPAS data.

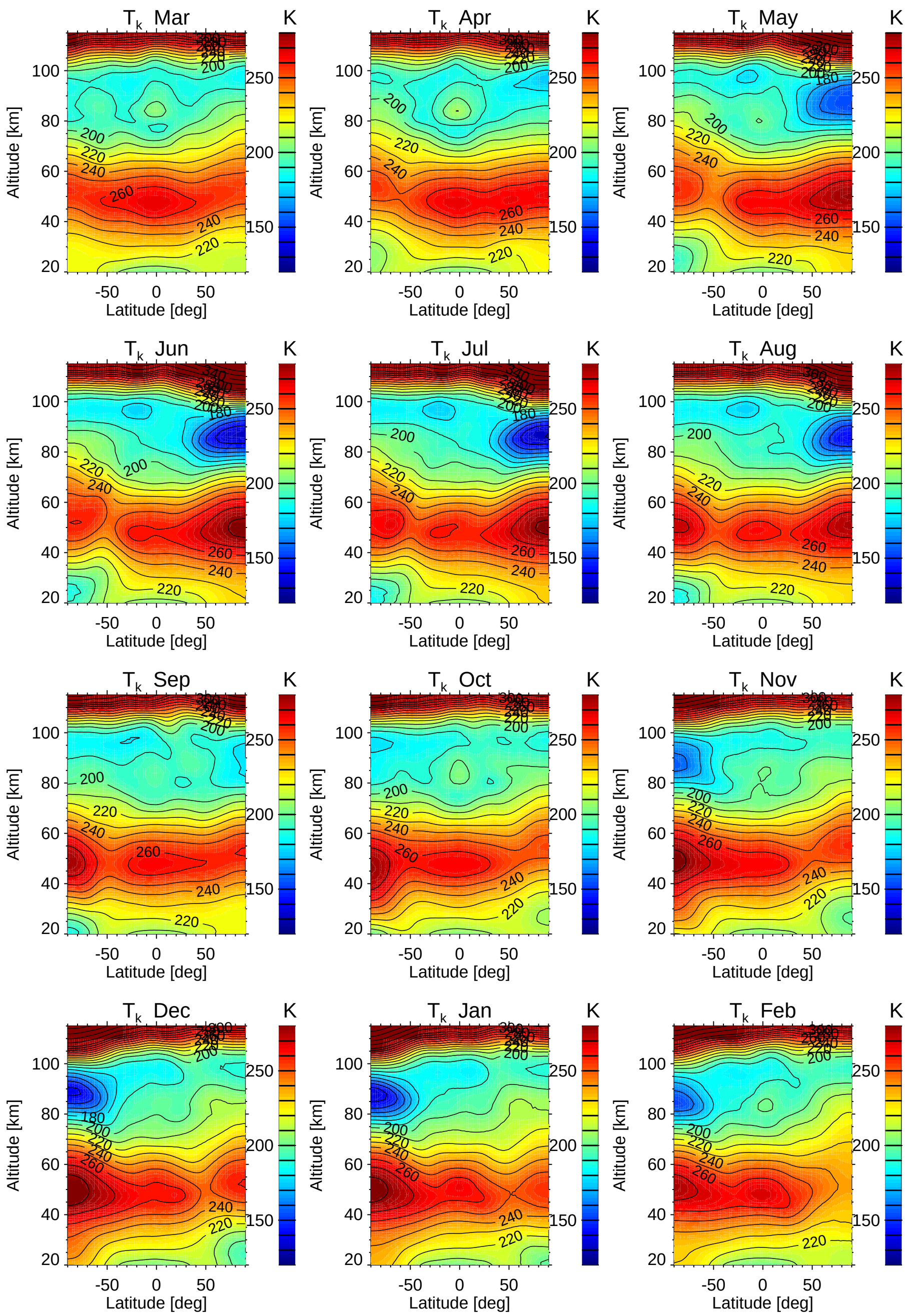

**Figure 1.** MIPAS V8 MA/UA composite monthly zonal mean temperature. Contour levels are indicated in the color bars (every 10 K up to ±300 K and then every 20 K.

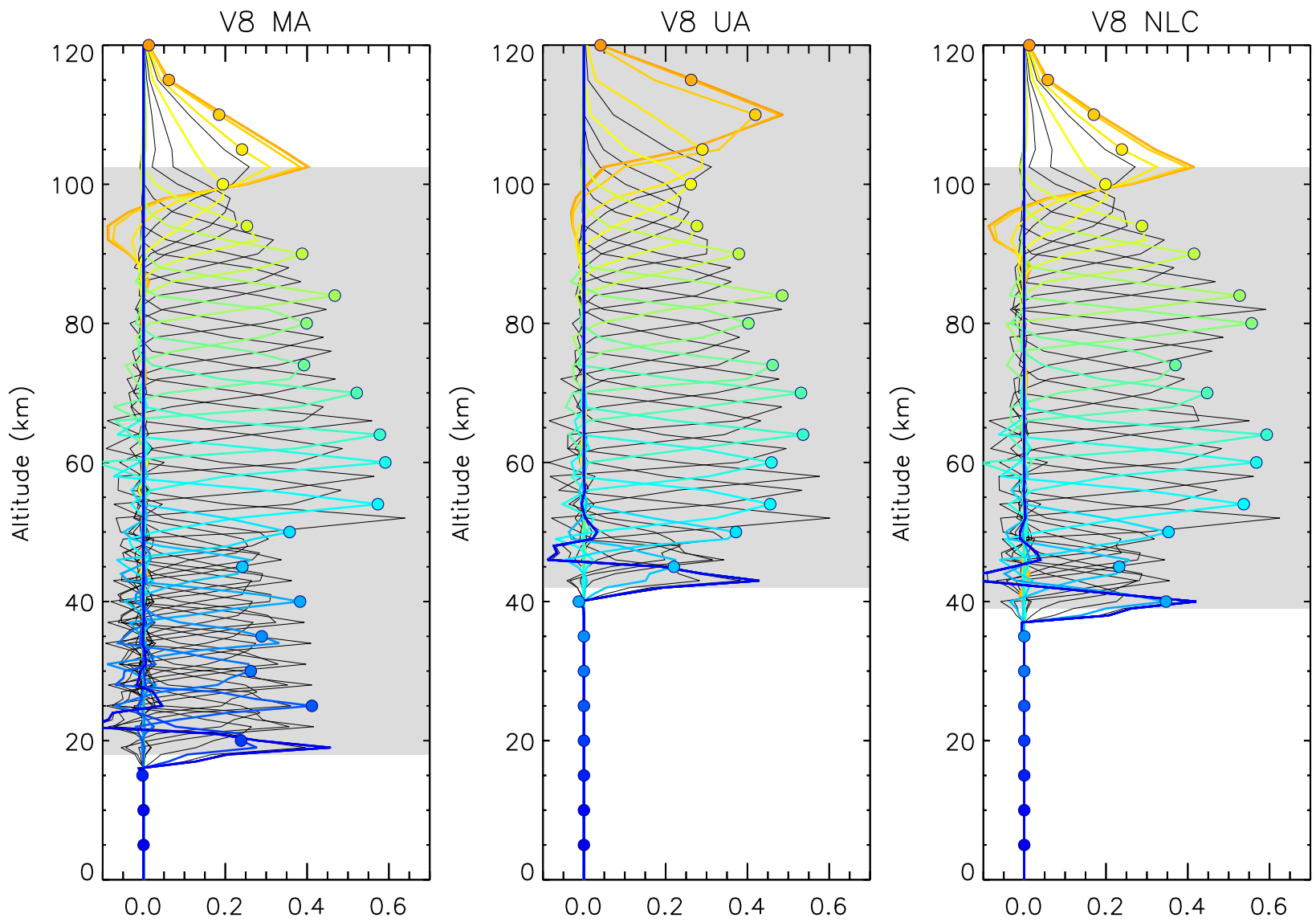

**Figure 2.** Example averaging kernels rows of the retrieved temperatures from MA (left), UA (center) and NLC (right) V8 measurements. Rows for retrievals from 5 to 120 km every 5 km are shown in color and the corresponding diagonal element values are shown with circles. Shaded areas indicate the altitude range of the measurements. The examples belong to measurements around 30°.

### 2.1 Retrieval performance

Figure 2 shows typical averaging kernel rows of the retrieved MA, UA and NLC V8 temperatures. There is temperature information from 18 to 110 km for MA measurements (starting at 39 km for NLC measurements) and from 42 to 115 km for UA measurements. The shifts between the peaks of the averaging kernels and the tangent heights observed at the highest altitudes (above 102 km tangent heights in MA and NLC, and above 110 km tangent heights in UA) emphasize the paramount importance of taking these averaging kernels into account when interpreting the data or when comparing temperatures with results from climate models or other temperature measurements. The averaging kernel matrix for each individual temperature profile is available with the data.

For each individual MIPAS temperature profile in our data, we provide the corresponding values of the vertical resolution, estimated as the full width at half maximum of the corresponding averaging kernel. Figure 3 shows the 2007–2012 composite of the vertical resolution of the V8 MA/UA temperature profiles. It is about 3 km at altitudes below 50 km, 3–5 km at 50–70 km, 4–6 km at 70–90 km, 6–10 km at 90–100 km and 8–11 km at 100–115 km. The vertical resolution of temperature shows no significant dependence on latitude or season, although it is slightly coarser at 90–100 km in the polar summer.

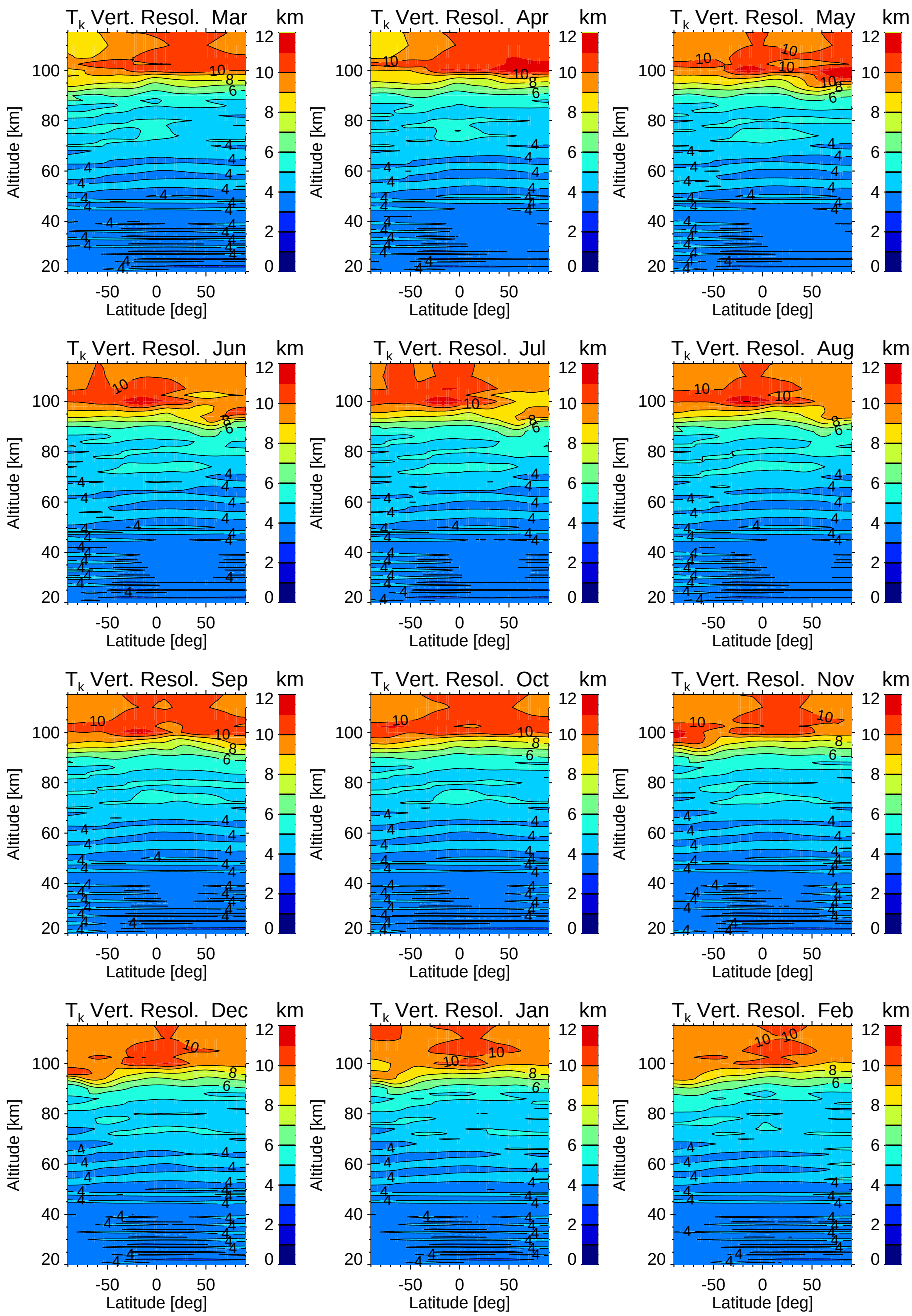

**Figure 3.** MIPAS V8 MA/UA composite monthly temperature vertical resolution. Contour levels are indicated in the color bars (every 1 km from up to ±10 km and then 12 km.

## 3 Error budget

Estimating the error budget for each individual MIPAS profile is computationally expensive. Rather than performing calculations for each profile, we have selected sets of uniformly distributed individual measurement profiles representative of typical atmospheric conditions, for which we estimated the temperature errors. We selected sets of about 30 geolocations for five latitude boxes (northern and southern poles, namely, 65°–90°; northern and southern mid-latitudes, namely, 40°–60°; and tropical, namely, 20°S–20°N), in the four seasons, and in day and night, resulting in 34 different atmospheric scenarios. Each component of the temperature error for each of these atmospheric conditions are then the mean of the corresponding component calculated for the individual geolocations within the set.

Table 3 summarizes the sources of temperature errors considered in our calculations and the associated uncertainties. In the following, we provide 1-$\sigma$ uncertainties. The assumed uncertainties are the same as those used by Kiefer et al. (2021) with the exception of the uncertainties related to the gain calibration, radiance offset and the non-LTE model. Except for the later, the assumptions made for their estimation are described in von Clarmann et al. (2022). We shortly justify the assumed uncertainties here:

−*Measurement noise*: The temperature noise error accounts for the propagation of measurement noise through the retrieval for single scans, calculated using the Level-1b wavelength dependent noise-equivalent- spectral-radiance (on average, 20 nW/ $(cm^2\, cm^{-1}\, sr)$ for MIPAS A band).

−*Radiance offset noise*: Although we have retrieved radiance offset together with the temperature, there is still a remaining random uncertainty due to the wavelength dependence of the deep space measurements used for the radiance offset calibration. We estimated the radiance offset uncertainty following the description in von Clarmann et al. (2022), which results in $3\,\mathrm{nW/(cm^2\,cm^{-1}\,sr)}$ for MIPAS Channel A working at reduced resolution.

−*Instrument line shape (ILS) uncertainties*: We followed the recommendations of Hase (2003) regarding uncertainties in the MIPAS instrument line shape (3%), characterized by the corresponding estimates of modulation loss through self-apodization.

−*Gain calibration uncertainties*: We estimated temperature errors separately for the systematic and for the random components of the gain calibration uncertainty. We assumed a systematic uncertainty of 1.1% (1-$\sigma$) for the gain calibration, re-scaled from the maximum scaling difference of 1.5% (2-$\sigma$) due to the calibration blackbody and the correction of the detector non-linearity (Kleinert et al., 2018). We have also considered a 0.2% gain calibration noise.

−*Shift uncertainties*: We estimated the uncertainty of the frequency shift to be 0.00029 $\mathrm{cm^{-1}}$ (see Kiefer et al., 2021), based on the deviation of the retrieved values from their linear fit along the wavelength.

−*$CO_2$ abundance uncertainty*: We assumed $CO_2$ volume mixing ratio uncertainties of 0.2% below 30 km, 0.5% at 40 km, 1% at 60–80 km, 10% at 90–100 km and 20% at 110 km. Below 60 km, these were based on uncertainties according to the IPCC Fifth Assessment Report, and also took into account $CO_2$ seasonal variability uncertainties. Above 60 km, we used the uncertainties estimated from previous WACCM $CO_2$ comparisons with ACE and SABER measurements (López-Puertas et al.,

**Table 3.** Temperature error sources and corresponding uncertainties considered in this work. The "Chief" column lists the nature of the dominating error component (sys: systematic; rdn: random). Error sources in each of the three sections of the table are listed following their relative contribution to temperature errors in decreasing order.

| Source | Chief | Uncertainty | Reference | Type♮ |
|---|---|---|---|---|
| *Measurement* | | | | |
| Noise | rdn | 15–33 nW/(cm$^2$sr cm$^{-1}$) | MIPAS Level1b data | G [8] |
| Radiance Offset noise | rdn | 3 nW/(cm$^2$sr cm$^{-1}$) | Kleinert et al. (2018) | G [5;13] |
| Instrument Line Shape | sys | 3% | Hase (2003) | P [7] |
| Gain calibration | sys | 1.2% (sys); 0.21% (rnd) | Kleinert et al. (2018) | P [7;12] |
| Spectral Shift | sys | 0.00029 cm$^{-1}$ | Kiefer et al. (2021) | P [7] |
| *Atmospheric constituents* | | | | |
| $CO_2$ | rdn | <1% below 60 km | 5$^{th}$ IPCC | P [7;10] |
| | sys | 2% at 80 km | López-Puertas et al. (2017) | |
| | sys | 10% at 90–100 km | " | |
| | sys | 30% at 120 km | " | |
| O♣ | sys | see Fig. 4 | López-Puertas et al. (2018) | P [7] |
| Interfering gases | rnd | see text | Preceeding V5 retrieval | G [6] |
| *Model parameters* | | | | |
| $CO_2(\nu_2)$ quenching by O ($k_O$)♣ | sys | 50% | García-Comas et al. (2012) | P [7] |
| $CO_2$ spectroscopy | sys | 1% (intensities) | M. Birk (pers. comm., 2020) | P [7] |
| | sys | 2% (p-broadening) | " | |
| | sys | 0.2 ($T_k$-dep. exponent) | " | |
| $CO_2(\nu_2)$ quenching by $N_2$ and $O_2$ ($k_{air}$)♣ | sys | 30% | García-Comas et al. (2012) | P [7] |
| $CO_2$ $\nu_2$-quanta exchange ($k_{vv}$)♣ | sys | 20% | " | P [7] |

♣ NLTE error sources. ♮ Error calculation propagation method: (G) generalized Gaussian error propagation in a matrix formalism; (P) via perturbation spectra. The numbers in brackets refer to the equation in von Clarmann et al. (2022), where the respective input uncertainty is applied.

2017). $CO_2$ uncertainties are positively correlated over small spatial or temporal scales and should be considered as source of systematic errors for localized comparisons with other instruments. In general, $CO_2$ uncertainties should be considered as systematic errors above 60 km.

275 −*$CO_2$ spectroscopy uncertainty*: Temperatures are retrieved from $CO_2$ emission lines using spectroscopic information from the 2016 HITRAN database. Uncertainties in the spectroscopy of the $CO_2$ lines are considered following recommendations by M. Birk (personal comm., 2020), namely, a 1% uncertainty in the intensity, a 2% uncertainty in the pressure-broadening coefficient and a 0.2 absolute uncertainty in the temperature-dependence exponent.

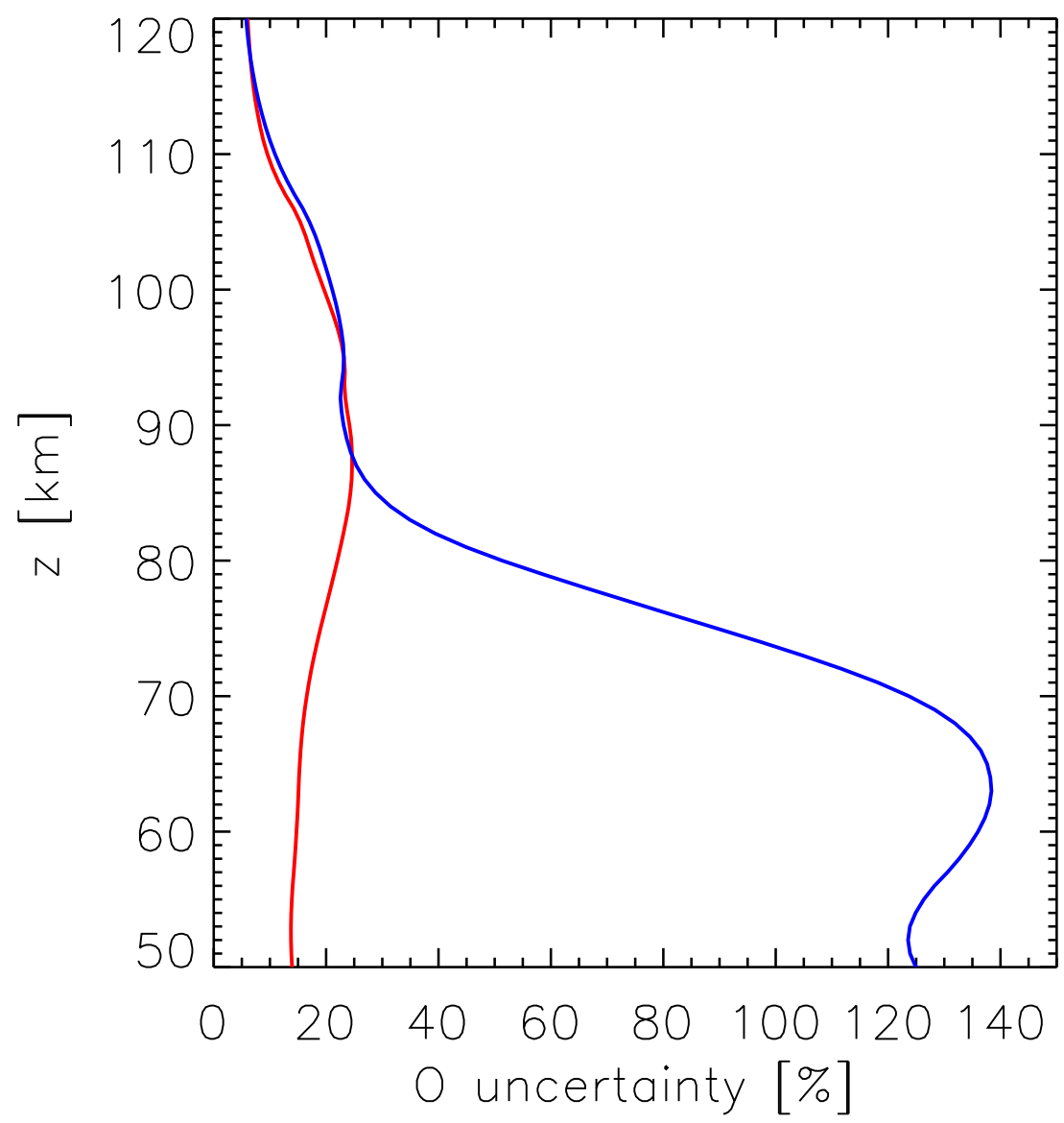

**Figure 4.** Estimated daytime (red) and nighttime (blue) uncertainties in atomic oxygen abundance averaged accross all seasons and latitudes.

−*Interfering gases uncertainty*: We have utilized the noise covariance information obtained from the preceeding MIPAS V5 retrievals, provided that the gases had been retrieved beforehand. In cases where the abundances interfering gases had not been previously retrieved, they were taken from our initial guess database (Kiefer et al., 2002), for which uncertainty information is often unavailable and was therefore estimated by educated-guesses. The contribution of these uncertainties to the temperature error is very small, rendering a more detailed assessment unnecessary.

−*Non-LTE modeling uncertainties*: The main source of error above the mid-mesosphere arises from uncertainties in the non-LTE modeling, which are mainly due to three collisional rates and the atomic oxygen abundance. Based on the considerations of García-Comas et al. (2012), we have assumed uncertainties of ±20% for the rate of vibrational exchange of $\nu_2$ quanta between $CO_2$ molecules, ±30% for the quenching of the $CO_2(\nu_2)$ states through collisions with $N_2$ and $O_2$, and ±50% for the quenching of the $CO_2(\nu_2)$ states by atomic oxygen.

The atomic oxygen below 95 km used in our temperature retrievals comes on average from MIPAS V5 daytime retrievals (we note that it is scaled for nighttime). Following López-Puertas et al. (2018) (see their Eqs. 5 and 6), we have estimated the atomic oxygen uncertainty by assuming their ozone abundance errors, a 10% uncertainty in the three-body reaction rate of $O_3$ formation and a 5% error in the ozone photodissociation. Above 97 km, we have estimated the uncertainty in the WACCM atomic oxygen used in our retrievals from comparisons with NRMLMSISE00 data. Estimates were made for 5 different latitude boxes, for the four seasons, and for day and night. On average, the uncertainties range from 25–30% at 85 km to 5% at 120 km (Fig. 4). Below this altitude, the nighttime uncertainties are significantly larger, reaching 120% at 70 km.

Following the recommendations of TUNER (Towards Unified Error Reporting; von Clarmann et al., 2020), we discuss systematic and random errors separately. This is useful because, for example, while the characterization of systematic errors is essential for the identification of measurement biases, they require less attention when evaluating atmospheric waves, atmospheric trends or instrumental drifts. von Clarmann et al. (2022) describe in great detail the methodology used in this work to estimate the contribution of the various error components to the temperature errors. We assume linear error propagation. Depending on the characteristics of each error source and the available information on its uncertainty, we chose between two different methods of error propagation: Gaussian or perturbation (see last column in Table 3). We applied Gaussian error propagation when the covariance matrix in the measurement domain was known (Eqs. 5 and 6 in von Clarmann et al., 2022), i.e., for measurement noise, noise impact on the radiance offset calibration measurement and noise-induced interfering gases uncertainties. We applied perturbation error propagation (Eq. 7 in von Clarmann et al., 2022) when Gaussian was not possible nor appropriate, i.e., for ILS, gain calibration, spectral shift, $CO_2$ abundance, $CO_2$ spectroscopy and non-LTE uncertainties. For error sources for which no specific values of their random uncertainty could be prescribed ($CO_2$ abundance and spectroscopy, non-LTE, ILS), we estimated their associated temperature random error from the dispersion of the effect of their uncertainty on the ensemble of profiles representing each atmospheric scenario.

Figures 5, 6 and 7, and the corresponding Tables A1–A4 in the Appendix, provide the overall random and systematic MIPAS V8 MA/UA temperature errors for representative polar (65°–90°), mid-latitude (40°–60°) and tropical (20°S–20°N) atmospheric scenarios, respectively, for selected seasons. The figures also show the contributions of the different error sources to the total error. Chiefly random errors are indicated with dashed lines and chiefly systematic errors are indicated with solid lines. This is only a selection of all the atmospheric scenarios we have considered. The Supplement of this manuscript contains a collection of the MIPAS MA/UA temperature error budgets for 34 representative atmospheric conditions for day and night of spring, summer, autumn, and winter conditions at polar latitudes, mid-latitudes and the tropics (see Table S0).

The MA/UA MIPAS temperature error budget behaves similarly as that for NOM at altitudes where MA/UA and NOM measurements overlap (below 68 km; see Kiefer et al., 2021). We focus our discussion here on the mesosphere and lower thermosphere. For the sake of completeness, we show the errors in the full altitude range covered by the MA/UA measurements.

### 3.1 Random errors

By far the largest contributor to the MIPAS temperature random error, which accounts for the temperature standard deviation, is the measurement noise. The noise error is less than 1 K at altitudes below 60 km, 1–3 K at 60–70 km, 3–5 K at 70–90 km, 6–8 K at 90–100 km, 8–12 K at 100–105 km and 12–20 K at 105–115 km (see Figs. 5−7 and the corresponding Tables A1–A4 in the Appendix).

The noise error does not exhibit a significant dependence with latitude or season, although it increases slightly around the polar summer mesopause (due to the lower temperatures there).

Less important than the measurement noise, but still non-negligible contributors to temperature random errors in the mesosphere and above are the following. The radiance offset noise typically results in random errors smaller than 0.5 K below 70 km, 1 K at 80 km, 2 K at 100 km and 5–7 K at 110–115 km. The $CO_2$ mixing ratios typically yield random errors smaller

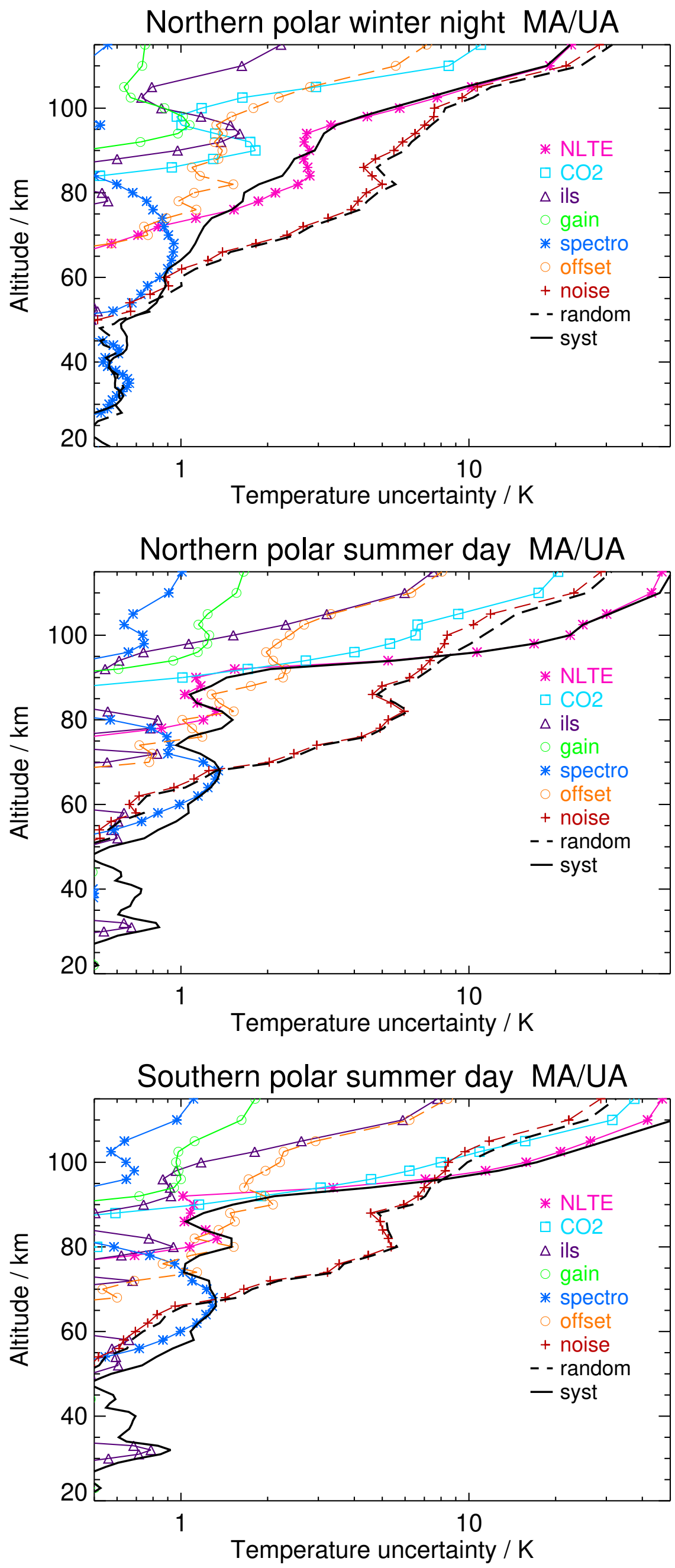

**Figure 5.** MIPAS V8 MA/UA temperature uncertainties for a few representative polar atmospheric scenarios. Chiefly random errors are indicated with dashed lines and chiefly systematic errors are indicated with solid lines. The corresponding error values are listed in Table A1 of the Appendix.

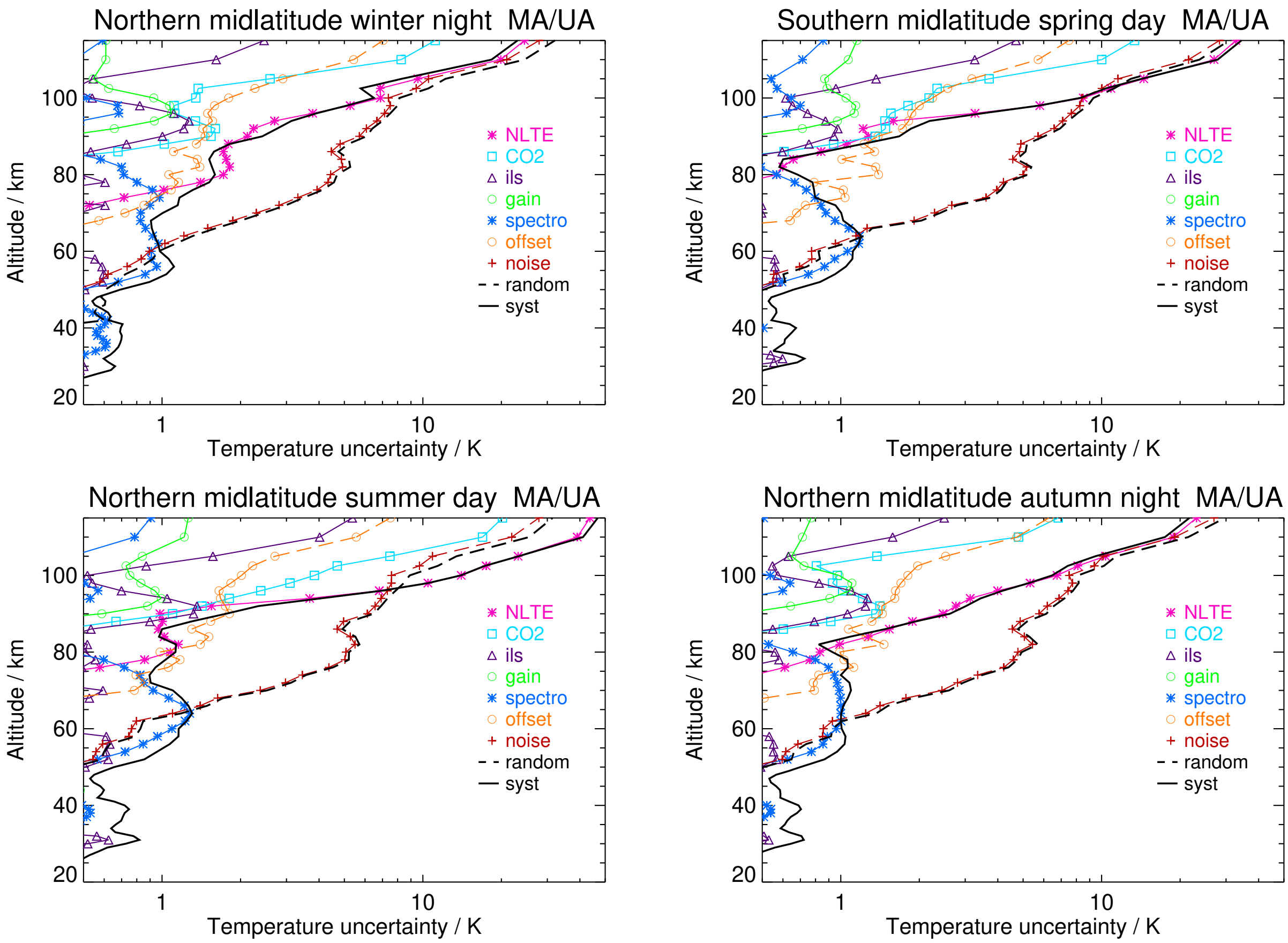

**Figure 6.** As Fig. 5 but for representative mid-latitude atmospheric scenarios. The corresponding error values are listed in Tables A2 and A3.

than 0.5 K below 90 km, 1–3 K at 105 km, 3–7 K at 115 km. The ILS uncertainties result in temperature errors less than 0.5 K below 80 km, 0.5–1 K at 80–105 km and 2 K at 115 km. Note that these random components are not shown individually in the figures.

Temperature random errors due to non-LTE model uncertainties are shown with dashed lines in Fig. 8. They are typically less than 1 K bellow 95 km (1.5 K for polar winter), 5 K at 105 km and 10 K at 115 km. The uncertainty in the $CO_2$-O quenching rate ($k_O$) is the primary responsible for these random errors, followed by uncertainties in the atomic oxygen abundance and, only under polar conditions around 80 km, the $CO_2$ quenching by $N_2$ and $O_2$ ($k_{air}$). According to our calculations, all other potential sources produce temperature random errors smaller than 0.5 K at all altitudes.

With respect to typical random errors in the tangent altitude correction, mainly arising from the instrumental noise, we found 50-60 m average values up to 60 km, decreasing linearly to values smaller than 20 m above 95 km (we recall here that we applied a very hard retrieval constraint towards ESA engineering tangent altitudes at 105 km).

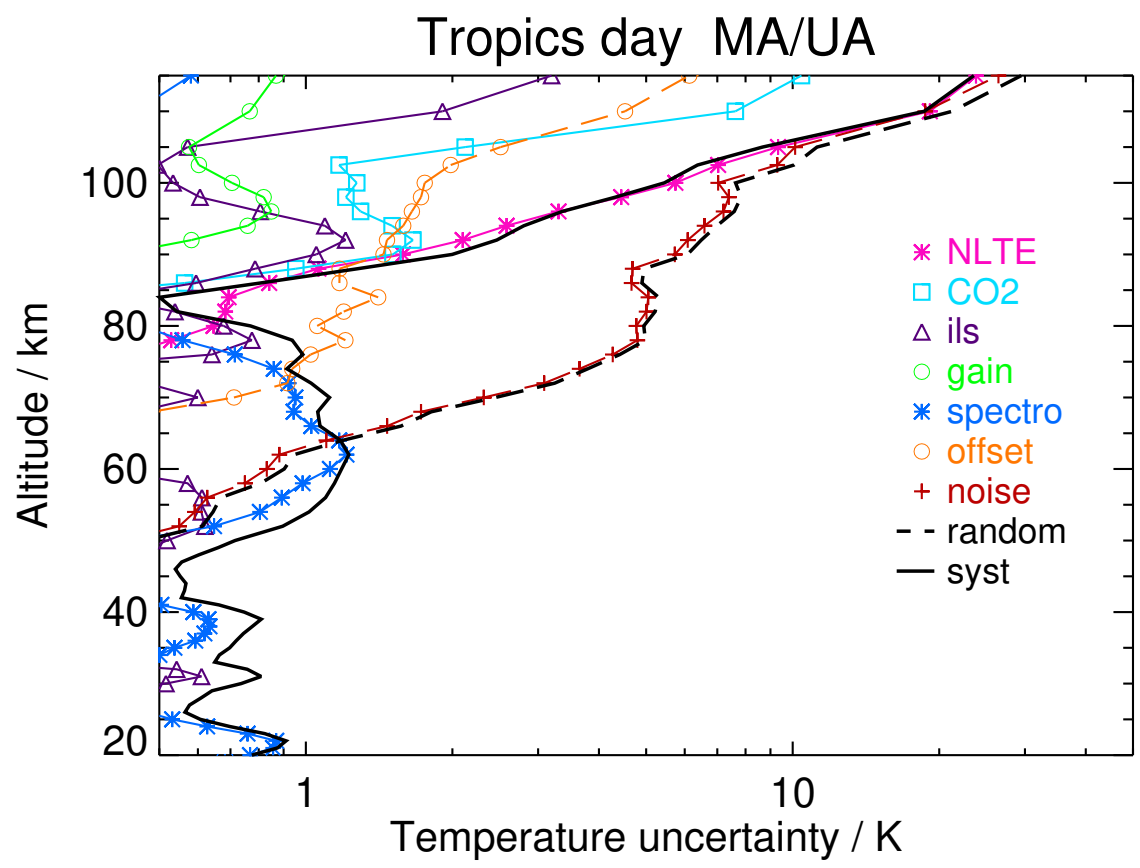

**Figure 7.** As Fig. 5 but for representative tropical atmospheric scenarios. The corresponding error values are listed in Table A4 of the Appendix.

### 3.2 Systematic errors

Figures 5−7 and the corresponding Tables A1–A4 in the Appendix also show the MIPAS temperature systematic errors, accounting for temperature biases. The main sources of systematic errors above the stratopause are the uncertainties in the $CO_2$ spectroscopic data (which govern the errors in the lower mesosphere, below 75 km); the non-LTE model parameters (which generally dominate above 75–80 km); the $CO_2$ abundance; the instrument line shape; and the gain calibration.

The spectroscopic information uncertainties typically lead to systematic errors smaller than 0.7 K below 55 km and around 1 K at 60–70 km. They are less than 0.5 K above 80 km, although they increase slightly in the lower thermosphere during the summer.

The overall non-LTE temperature systematic errors, together with the contributions of each non-LTE model parameter uncertainty, are shown as solid lines in Fig. 8. They are highly dependent on the atmospheric conditions. Overall, non-LTE systematic errors are smaller than 0.5 K typically below 75–80 km and below 65–70 km for the polar and mid-latitude winters. The lower altitude for the latter is due to the contribution of the $k_{air}$ uncertainty. These errors generally increase to 1 K at 80–90 km in the mid-latitude and polar summers, due to the contribution of $k_{air}$ and $k_{vv}$ uncertainties, and to 2 K in the polar winter, mainly due to the contribution of the $k_O$ uncertainty.

At 95 km and above, the uncertainty in $k_O$ clearly dominates the non-LTE systematic error and thus the total systematic error (Figs. 5−7). Non-LTE errors and, thus, total systematic error, generally reach 3 K at 95 km, 6–8 K at 100 km, 10–20 K at 105 km and 20–30 K at 115 km. For the mid-latitude and polar summers, they are somewhat larger (4–5 K at 95 km, 10–15 K at 100 km, 20–30 K at 105 km and 40–50 K at 115 km), due to the enhanced non-LTE effect for the steep temperature gradient and large temperatures for these conditions. The error due to the atomic oxygen abundance uncertainty displays contributions larger than 1 K above 100 km (95 km in the polar summer) but is significantly smaller than that due to $k_O$. In fact, we estimate

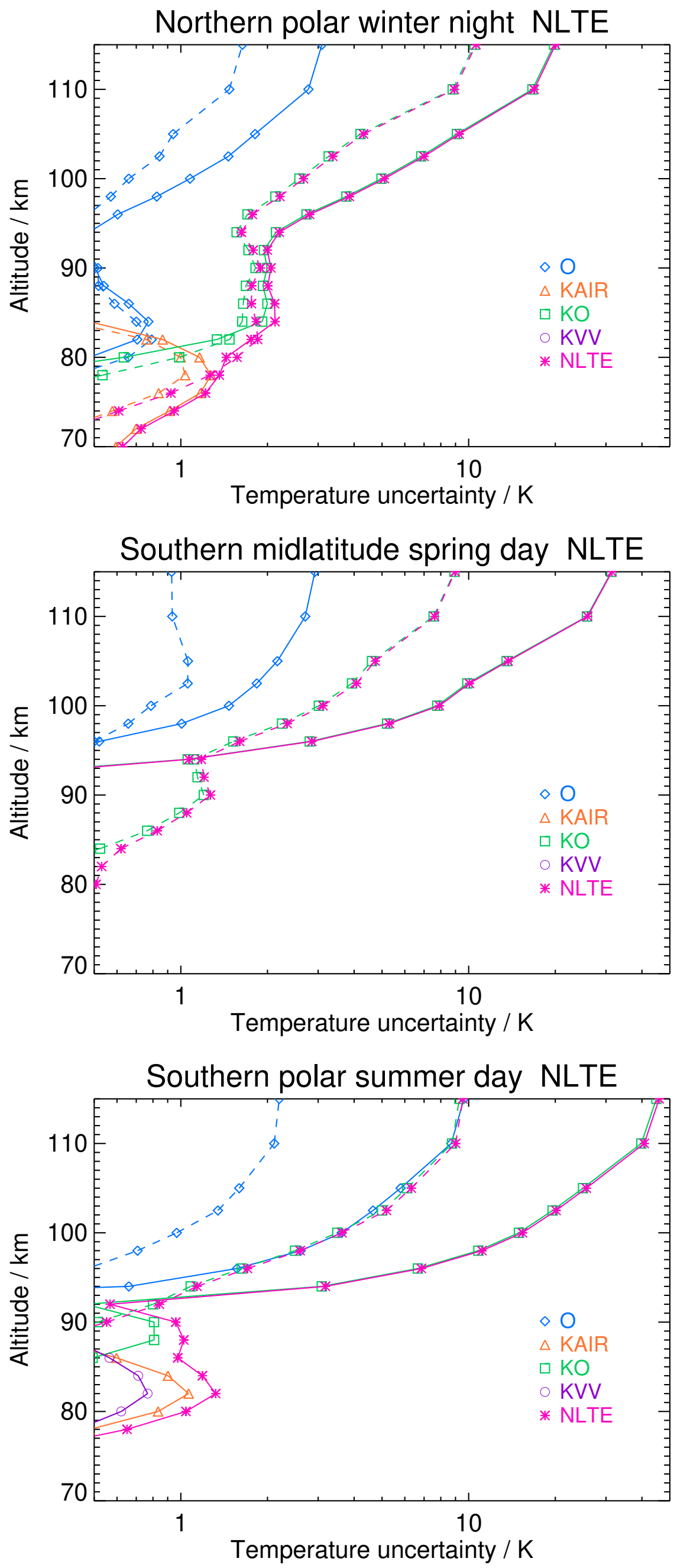

**Figure 8.** MIPAS V8 MA/UA temperature systematic (solid line) and random (dashed lines) errors due to non-LTE model uncertainties, i.e., uncertainties in atomic oxygen abundance (blue), rates of $CO_2(\nu_2)$ quenching by O (green) and by $N_2$ and $O_2$ (orange), and rate of $\nu_2$ exchange between $CO_2$ molecules (violet).

that, the second largest contributor to the temperature systematic error above 90–95 km, after $k_O$ but before the atomic oxygen abundance uncertainty, is the uncertainty in the $CO_2$ abundance. It has little effect ($< 1$ K) at lower altitudes but yields typical errors of 1–2 K at 95–105 km and 7–10 K at 110–115 km, and errors of 2–7 K at 95–105 km and around 20 K at 115 km in the mid-latitude and polar summers.

The ILS uncertainty has some effect at mesospheric altitudes but is generally smaller than 0.5–0.7 K below 80 km, i.e., smaller than that coming from the $CO_2$ spectroscopy. It is comparable to that of the $CO_2$ or the $k_O$ uncertainties at 85–90 km but always becomes smaller than both above 95 km.

The contribution of the radiance gain uncertainty is even smaller than that of the ILS. Other potential sources of systematic errors we considered, as the spectral shift or interference by $N_2O_5$, cause errors smaller than 0.5 K at all altitudes. They are listed in Table 3 but their effects are not visible in the figures.

As for the typical systematic errors in the tangent altitude correction, mainly arising from the $CO_2$ spectroscopic uncertainties, they are on average 250 m at 20 km and 200 m from 40 to 60 km, decreasing linearly to 100 m at 80 km and smaller than 50 m above 90 km.

## 4 Temperature differences between the current V8 and the previous V5 data versions

The improvements in this version of temperature retrievals (V8) over the previous version (V5) are listed in Sect. 2. The resulting differences between the two versions are dominated by the changes in the atomic oxygen and carbon dioxide abundances and in the a priori temperature gradients. Figure 9 shows the differences in the retrieved temperatures produced by these upgrades. We have limited the figure to altitudes above 70 km because the differences below are less than 0.5 K.

As we show in Fig. 9, the change in V8 with respect to V5 at tropical and mid-latitudes has a vertically oscillating structure up to about 105 km, with alternating positive and negative differences, somewhat more pronounced during March, April and May. The differences are negative and smaller than 1 K (in absolute value) below 85 km, positive and smaller than 3 K between 85 and 95 km, and negative and also smaller than 3 K (in absolute value) between 95 and 105 km. This is a manifestation of the better representation of the tides in the MIPAS V8 temperatures relative to previous versions. In addition, although not shown here, the effects are slightly different during the day than at night. This day-night differential behavior of the oscillating differences additionally results in slightly enhanced diurnal migrating tide amplitudes derived from version 8 data below 105 km (1–5 K larger depending on altitude and season; see the am-pm differences in Fig. 19 in Funke et al. (2023), using this dataset below 115 km).

During the polar summer, the structure of the differences between 75 and 100 km is caused by minor decreases in height and temperature of the V8 mesopause in comparison to the V5 mesopause. In contrast, during the polar winter, the V8 mesopause is generally slightly elevated and only sometimes cooler, leading to a distinct behavior of the differences.

Above 105 km, version 8 delivers higher temperatures than version 5 under all atmospheric conditions, resulting in differences that grow monotonically from about 5 K at 105 km to 25–30 K at 115 km, i.e., the temperature vertical gradient in the lower thermosphere is larger in V8. At low-to-mid latitudes, the change from V5 to V8 is also not the same between day and

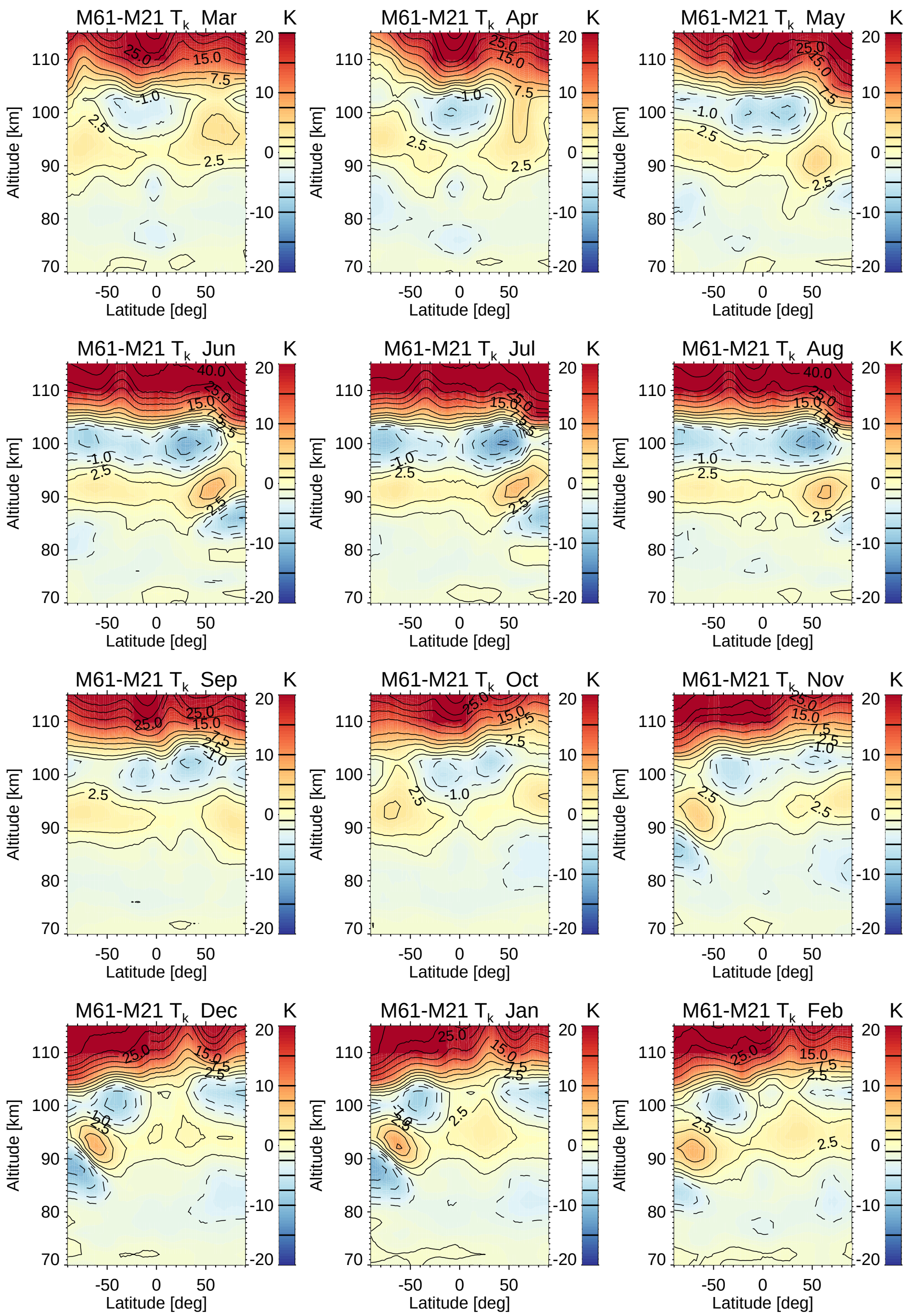

**Figure 9.** MIPAS V8 and V5 MA/UA zonal mean temperature differences. Contour levels are indicated in the color bars (±1 K, ±2.5 K, ±5 K, ±7.5 K, , ±10 K and every ±5 K above that value).

night (not shown). This is of particular interest for the analysis of tides in the lower thermosphere and has an impact on the findings presented by García-Comas et al. (2016) regarding the latitudinal structure of the diurnal migrating tide at 105-115 km, as depicted in Fig. 19 in Funke et al. (2023).

Note that the change from version 5 to version 8 in polar winter in MA/UA is not analogous to NOM measurements, for which the improvements are significant (Kiefer et al., 2021). This is because V8 NOM temperatures solved a known problem in V5 during elevated stratopause episodes by including more realistic a priori temperatures above 60 km. Unlike V5 NOM temperatures, earlier MA/UA temperature versions were not affected by this problem, because altitudes above 70 km were covered by measurements. Thus, the differences between V8 and V5 in polar winter are not as pronounced in MA/UA as in NOM

## 5 Consistency with NOM mode measurements

As mentioned above, MIPAS made nominal measurements from 6 to 70 km (NOMinal mode). Versions of IMK/IAA NOM retrievals previous to V8 did not use the same microwindow set as the MA/UA/NLC retrievals (five of the 14 microwindows listed in Table 1 were not included). Furthermore, previous NOM retrievals did not consider non-LTE conditions, not even in an approximate manner. As described in Sect. 2, the NOM V8R_T_261 retrievals use the same microwindows as MA/UA/NLC V8R_T_m61 and, while not the full non-LTE treatment, at least a simplified one (see Kiefer et al., 2021). Both upgrades have led to an improvement in the agreement between the V8 MA/UA/NLC and the NOM datasets as compared to previous versions.

Here we compare 2007-2012 MA/UA V8R_T_m61 and NOM V8R_T_261 temperature fields in the altitude range where they overlap to check for their consistency. Figure 10 shows 2007-2012 composites of MIPAS V8 MA mode and NOM mode temperatures for the four seasons, and their differences.

MA/UA and NOM temperatures are within about 0.5 K below 70 km at all latitudes and seasons. Exceptions occur only for some localized areas in the polar latitudes in specific seasons, namely, in the northern polar winter and in the northern polar latitudes from September to November (SON) (0.75–1.5 K differences), and in the lower stratosphere at southern polar latitudes also from September to November (2–3 K differences). Note that the MIPAS NOM and MA measurements are not simultaneous, so part of these small differences can be explained by the observational mismatch. These comparisons show that, in general, MIPAS V8 MA/UA and NOM temperatures can be safely combined without special consideration.

## 6 Comparison with SABER

García-Comas et al. (2012) and García-Comas et al. (2014) showed comparisons of previous versions of temperature retrievals from MIPAS MA, UA and NLC 5.02/5.06 spectra with measurements from several instruments. In general, MIPAS V5 temperatures showed deviations from the other instruments within 1 K below 50 km, 2 K at 50-–80 km, 4 K at 80–95 km and 5 K from 95–105 km. Exceptionally, during high latitude summers, deviations from other instruments were less than 5 K between

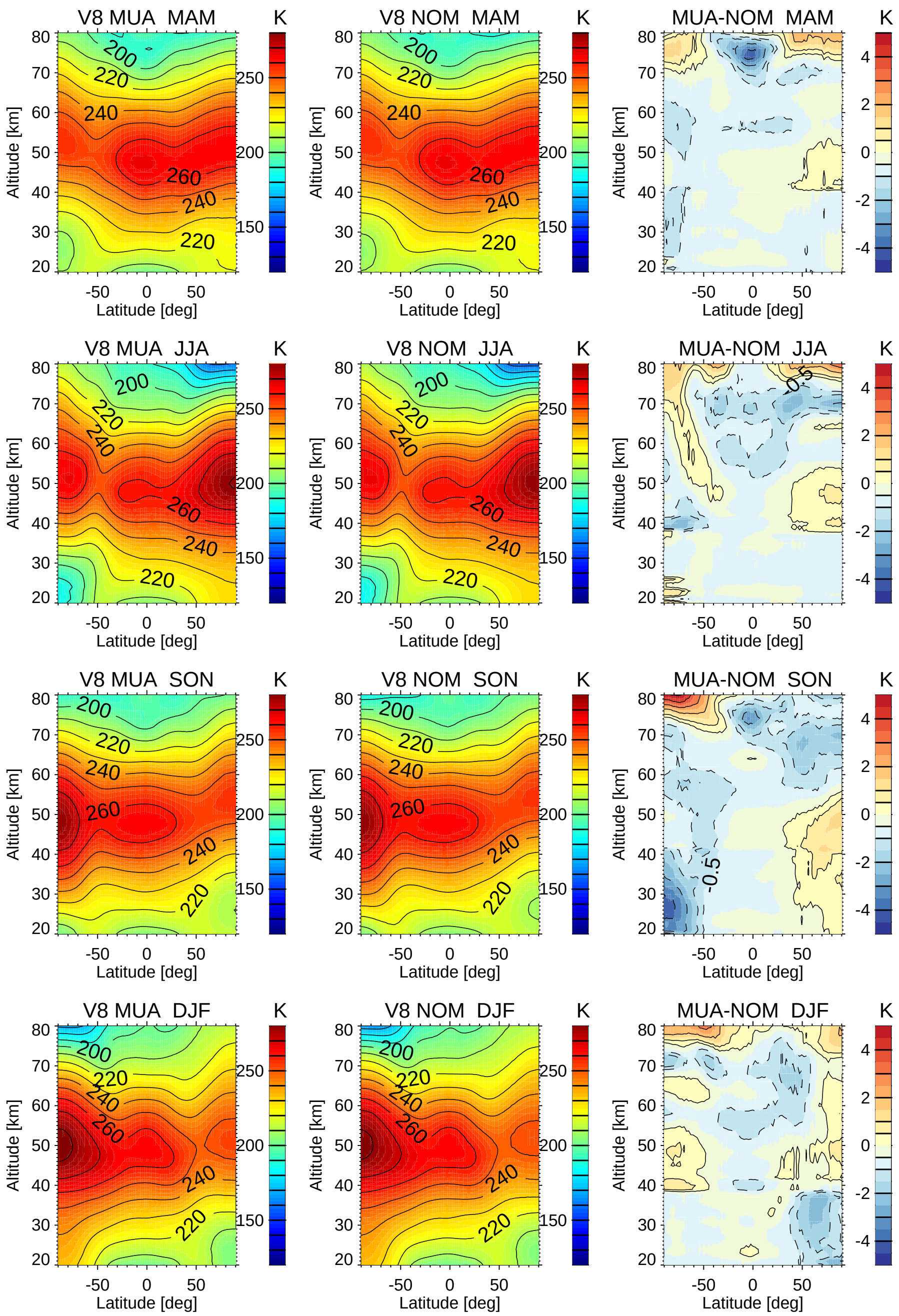

**Figure 10.** Comparison between 2007-2012 composites of MIPAS V8 MA/UA (left column), and V8 NOM (middle column) zonal mean temperatures for the four seasons. The differences are shown in the right column (contour levels are ±0.5, ±1, ±2, ±3, ±4, ±5).

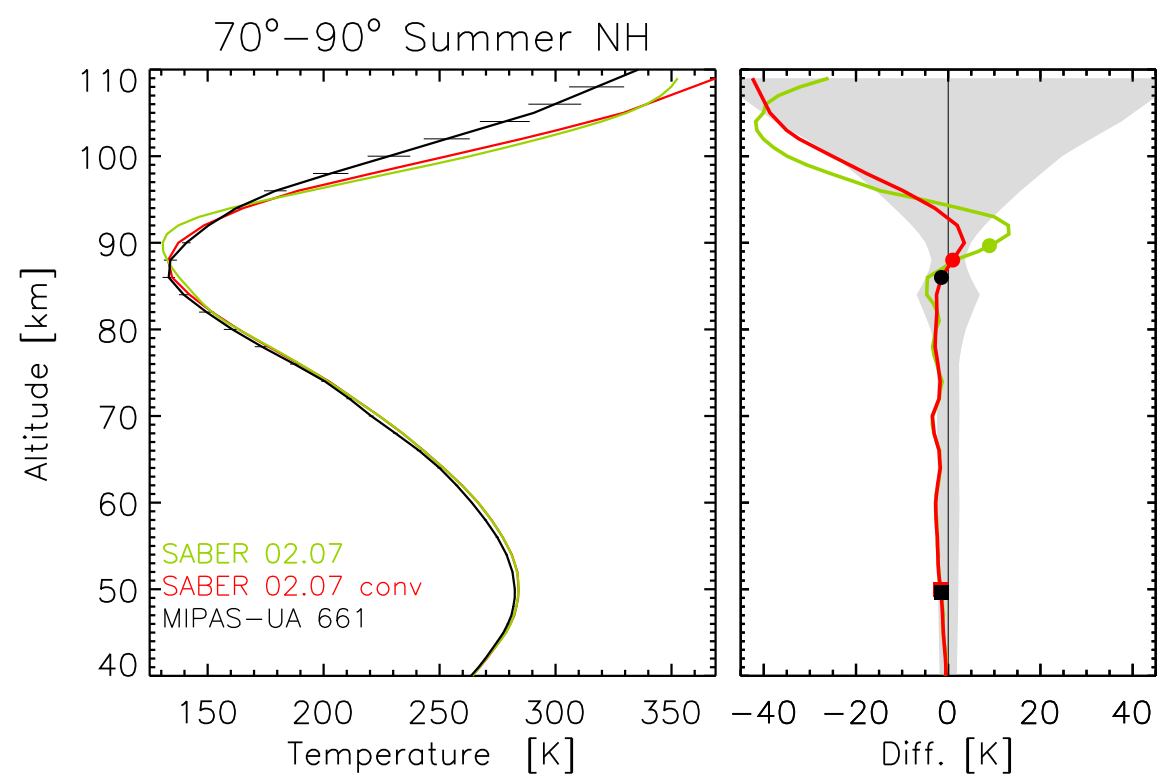

**Figure 11.** Effect of applying MIPAS averaging kernels to SABER co-located profiles to match MIPAS vertical resolution for the polar summer at 70°N–90°N (black: MIPAS V8 UA; red: SABER smoothed; green: SABER un-smoothed). The right plot shows MIPAS−SABER temperature differences. The filled squares and circles locate the stratopause and the mesopause, respectively. The shaded area shows MIPAS and SABER combined systematic errors.

65–80 km and increased to 5–10 K around the summer mesopause. MIPAS usually displayed larger vertical gradients in the thermosphere than the other instruments.

Here we compare MIPAS V8 MA/UA temperatures with NASA's Sounding of the Atmosphere using Broadband Emission Radiometry (SABER) data version 2.0 (Remsberg et al., 2008; García-Comas et al., 2008). As for MIPAS, SABER temperatures are retrieved from measurements at 15 $\mu$m considering explicit non-LTE calculations for each limb scan, although it uses a broadband channel and hence is sensitive to many $CO_2$ 15 $\mu$m bands. SABER temperature profiles are publicly available from 20 km to 110 km and have a 2 km vertical resolution. We selected MIPAS and SABER co-located profiles within 1000 km and 2 hours. To avoid differences emanating from the coarser vertical resolution of MIPAS, especially above the stratopause or the use of a priori information, for a given pair of MIPAS and SABER profiles, we use the corresponding individual MIPAS averaging kernel matrix ($\mathbf{A}_{\text{MIP}}$) and a priori ($\mathbf{t}_{a,\text{MIP}}$) to smooth the colocated SABER profile ($\mathbf{t}_{\text{MIP}}$), that is to say, $\mathbf{t}_{\text{SAB},smoo} = \mathbf{t}_{a,\text{MIP}} + \mathbf{A}_{\text{MIP}}(\mathbf{t}_{\text{SAB}} - \mathbf{t}_{a,\text{MIP}})$.

Figure 11 shows the composite effect of such a smoothing on SABER profiles in the most extreme case, northern polar summer conditions. This composite is the mean difference between collocated pairs of SABER and MIPAS measurements within 70°N–90°N on June, July and August from 2005 to 2012. Below 80 km, the composite SABER smoothed profile is not significantly different from the unsmoothed one. However, the SABER smoothed mesopause is 2 km lower and 3.5 K warmer than its unsmoothed counterpart, leading to a significantly better agreement with MIPAS. The lower thermospheric temperatures are also closer to MIPAS when the smoothing is applied. This highlights not only the well-known importance of smoothing in terms of profile-to-profile comparisons but also for the scientific interpretation of the data, especially since the smoothing effect exhibits temporal and spatial variations.

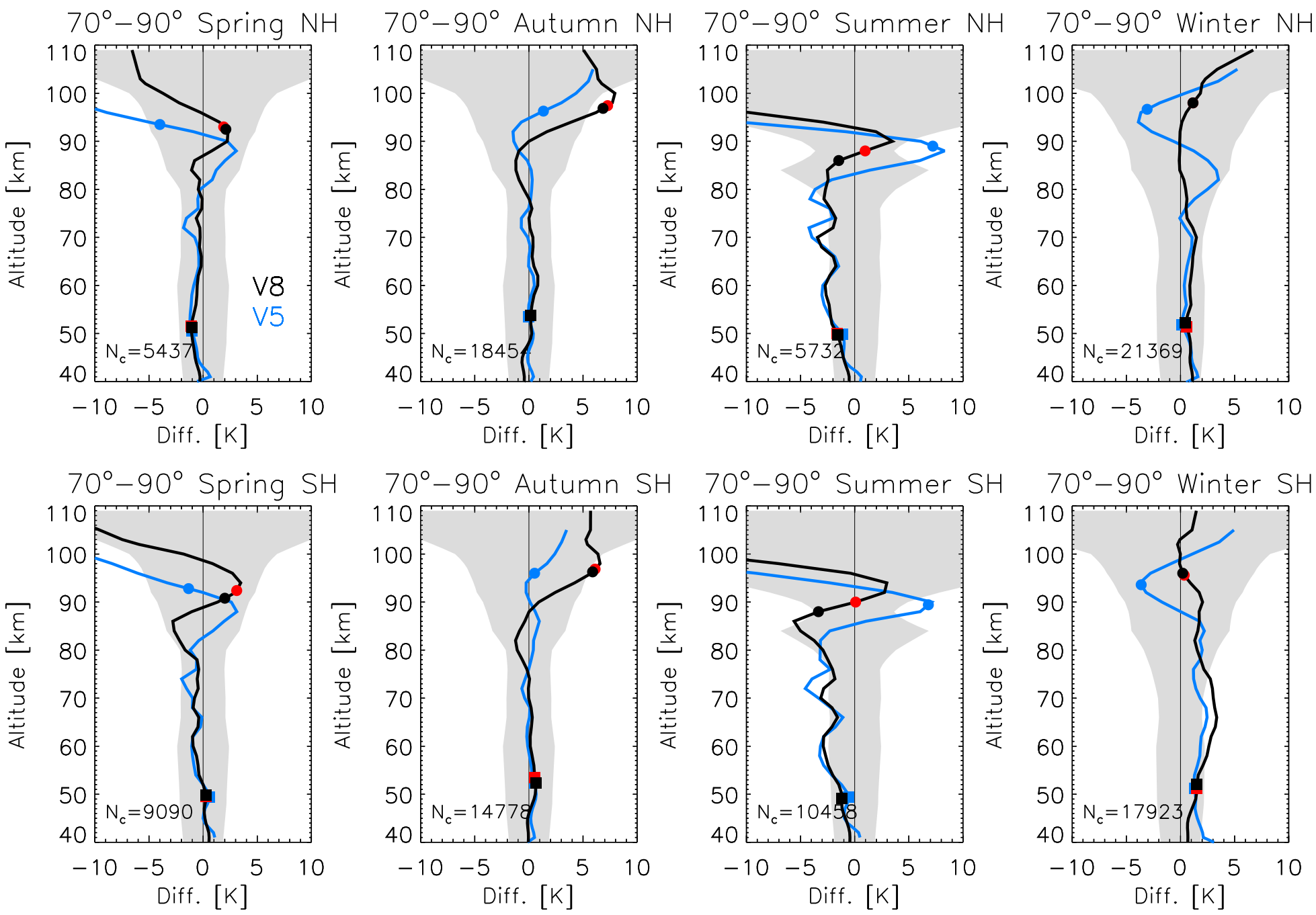

**Figure 12.** Co-located MIPAS-SABER differences averaged for northern (upper panels) and southern (lower panels) polar latitudes (70°–90°). Black: MIPAS V8; blue: MIPAS V5. Shaded areas show MIPAS and SABER combined systematic errors. The filled squares and circles on the lines indicate the mean altitude of the stratopause and the mesopause, respectively, also for SABER (red).

We have performed similar comparisons for five latitude boxes and the four seasons from 2005 to 2012. Figures 12-15 show the MIPAS-SABER differences averaged for northern and southern 70°–90°, 50°–70°, 30°–50° and 10°–30° latitude boxes, respectively. Each figure shows results for the Northern and Nouthern Hemispheres separately. Differences with SABER are shown for MIPAS V5 temperatures to examine the improvements in the V8 version presented here. The figures also depict MIPAS and SABER combined systematic temperature errors for the corresponding atmospheric conditions. These combined errors account for existing correlations between MIPAS and SABER non-LTE model uncertainties, specifically the quenching rates of $CO_2(\nu_2)$ by O, $N_2$ and $O_2$, which are set to the same values in the temperature retrievals of both instruments.

Next, we list a selection of the most salient results:

- As a rule of thumb, MIPAS V8 and SABER temperatures are in excellent agreement below 90 km. Temperature differences are typically within 1.5 K at these altitudes and within the combined systematic errors. In general, comparisons with V8 improved over those with V5 at 80–90 km.

- There are exceptions to this rule: during summer at 70°–90° in the mesosphere (3 K, MIPAS being colder), with an improvement in the upper mesosphere with respect to V5; during winter at 50°–70° in both hemispheres and 70°–90°

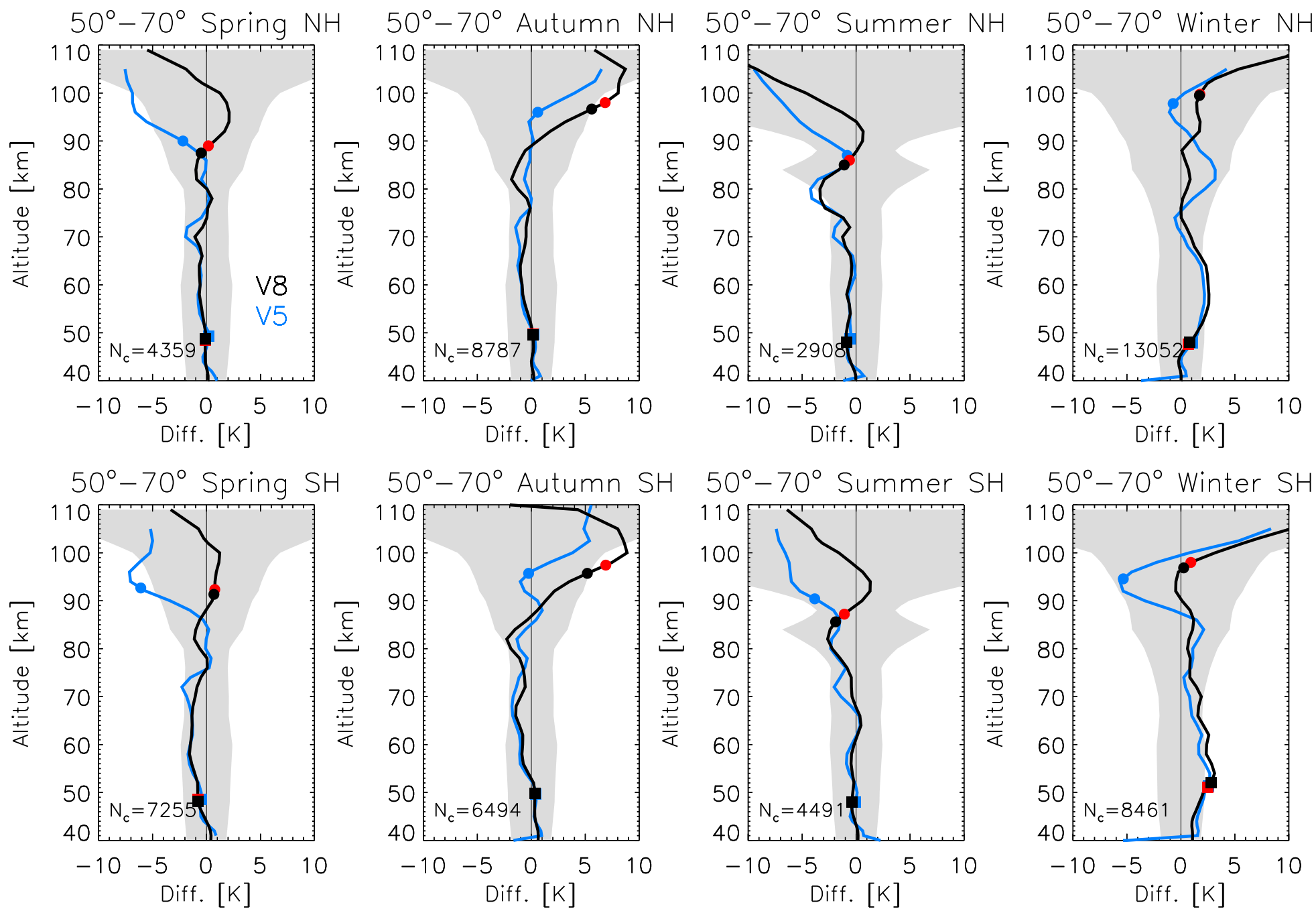

**Figure 13.** As Fig. 12 but for the 50°N–70°N (upper panels) and 50°S–70°S (lower panels) latitude boxes.

in the Northern Hemisphere only in the lower mesosphere (3 K, MIPAS being warmer), thanks to the improvements at these latitudes at 80–90 km with respect to V5; and, during summer at 80 km and winter in the upper stratosphere (2 K, MIPAS being warmer), as for V5.

– The changes in MIPAS temperatures at 80–90 km from V8 to V5 at 70°–90° have reduced the differences with SABER from 6–7 K to 3–4 K in the summer and from 3–4 K to 0–1.5 K in the winter. This has further reduced the hemispheric asymmetry of the differences.

– V8 temperatures are higher than V5 at all altitudes above 90 km, except above 100 km in the polar winters. This leads to improvements in the comparisons with SABER in many cases but not always. V8 comparisons improve significantly in summer at latitudes equatorward of 70° (V8 and SABER differences are within 3–4 K at 90–100 km, 4–6 K at 100–110 km in the Southern Hemisphere and 5–10 K at 100–110 km in the Northern Hemisphere). The improvement is not sufficient to bring the MIPAS and SABER polar summer temperatures closer than 10 K above 95 km. However, the differences generally fall within the range of the combined retrieval error.

– The typically higher temperatures above 90 km in V8 also lead to improvements at 90–100 km in the winter, particularly in the Southern Hemisphere and also at high latitudes (differences within 3–4 K). They also further lead to V8 improve-

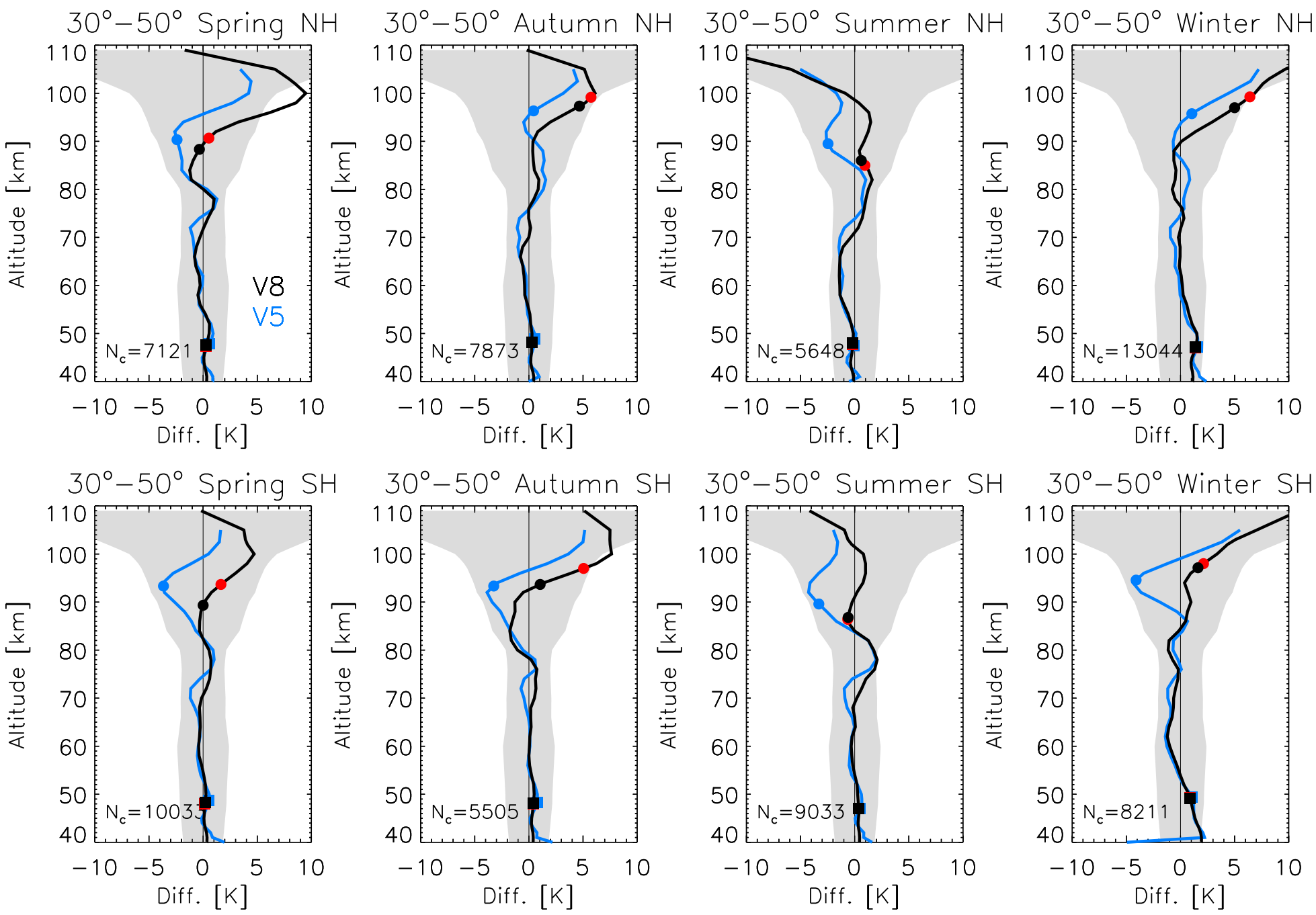

**Figure 14.** As Fig. 12 but for but for northern (30°N–50°N; upper panels) and southern (30°S–50°S; lower panels) mid-latitudes.

ments in spring at latitudes poleward of 30° in the Southern Hemisphere (differences within 3–4 K) and poleward of 50° in the Northern Hemisphere (differences within 3–5 K).

- However, the comparisons of MIPAS V8 with SABER deteriorate at 100–110 km elsewhere in spring, and also everywhere in autumn and winter (except at latitudes of 70°–90°). In these cases, the differences increase from 1–6 K in V5 to 4–9 K in V8. With the exception of the polar autumn at 100 km, they still fall within the estimated combined errors.
- At altitudes ranging from 100 to 110 km and latitudes ranging from 70° to 90° during the winter, V8 differences with SABER are less than 1 K in the Southern Hemisphere and less than 6 K an the Northern Hemisphere.
- There are slight changes in the MIPAS–SABER differences from one hemisphere to the other, but it is difficult to glimpse any systematic behavior. The most striking examples of asymmetry are the summers at latitudes poleward of 30° above 95 km.

We have also compared MIPAS V8 temperature time series with the smoothed co-located SABER data (i.e., averaging kernels and a priori information applied). Figure 16 shows such comparisons for northern polar and mid- latitudes for selected altitudes (note the different scales in the abscissas). We started the time series at the end of 2007 because MIPAS data are not available before for some seasons.

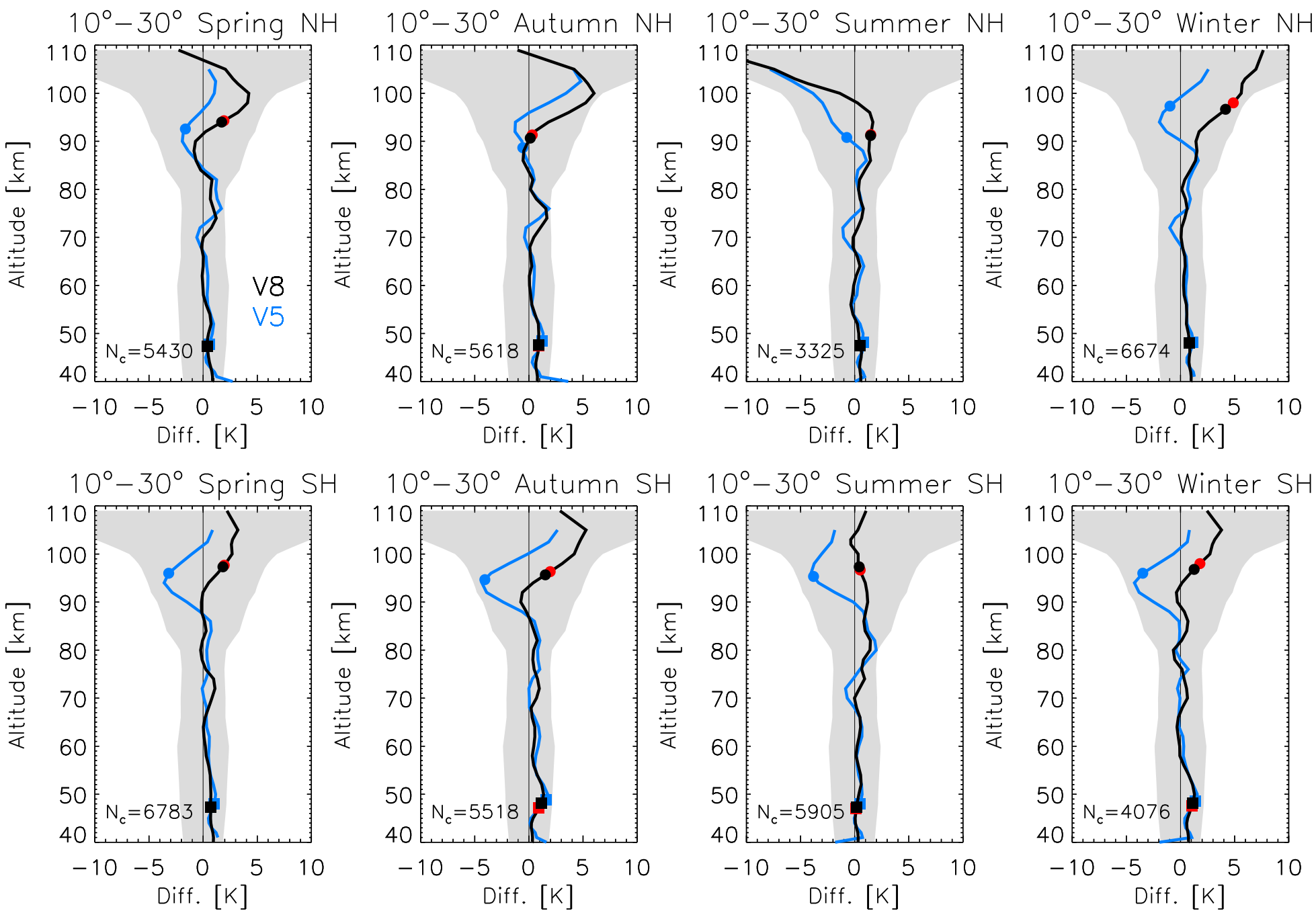

**Figure 15.** As Fig. 12 but for but for northern (10°N–30°N; upper panels) and southern (10°S–30°S; lower panels) tropical latitudes.

The agreement at the lowermost altitude shown (85 km) is remarkable. Temperatures agree within 1–2 K each season and each year. Both instruments exhibit similar peak-to-peak amplitudes (60–80 K in the poles; 20–25 K at mid-latitudes), which vary similarly from year to year. At polar latitudes, the agreement at 95 km is somewhat worse (peak-to-peak amplitudes are 5 K larger for MIPAS). However, the seasonal behavior of the differences is more or less consistent from year to year. At 95 km in mid-latitudes, MIPAS temperatures are always 2–3 K higher than SABER, leading to a similar seasonal variation but shifted.

At 100 km, MIPAS northern polar temperatures are lower than SABER in all winters and higher in all summers and winter-summer differences change very slightly from year to year. This means that while MIPAS provides peak-to-peak amplitudes between 30 and 45 K, SABER provides values between 50 and 75 K. A similar behavior is also observed at mid-latitudes but with smaller amplitudes (10 K for MIPAS and 15 K for SABER), both instruments showing a rather small year-to-year variation.

At the uppermost altitude shown in Fig. 16 (108 km), MIPAS polar winter temperatures are 35–40 K lower than those from SABER, while they are 5–10 K warmer in the summer. This leads to peak-to-peak amplitudes of 90–100 K for MIPAS but 120–140 K for SABER. However, the difference between the instruments here does not vary from year to year. For mid-latitudes, the behavior is similar but less pronounced, with MIPAS yielding 20–25 K amplitudes and SABER 40–50 K amplitudes.

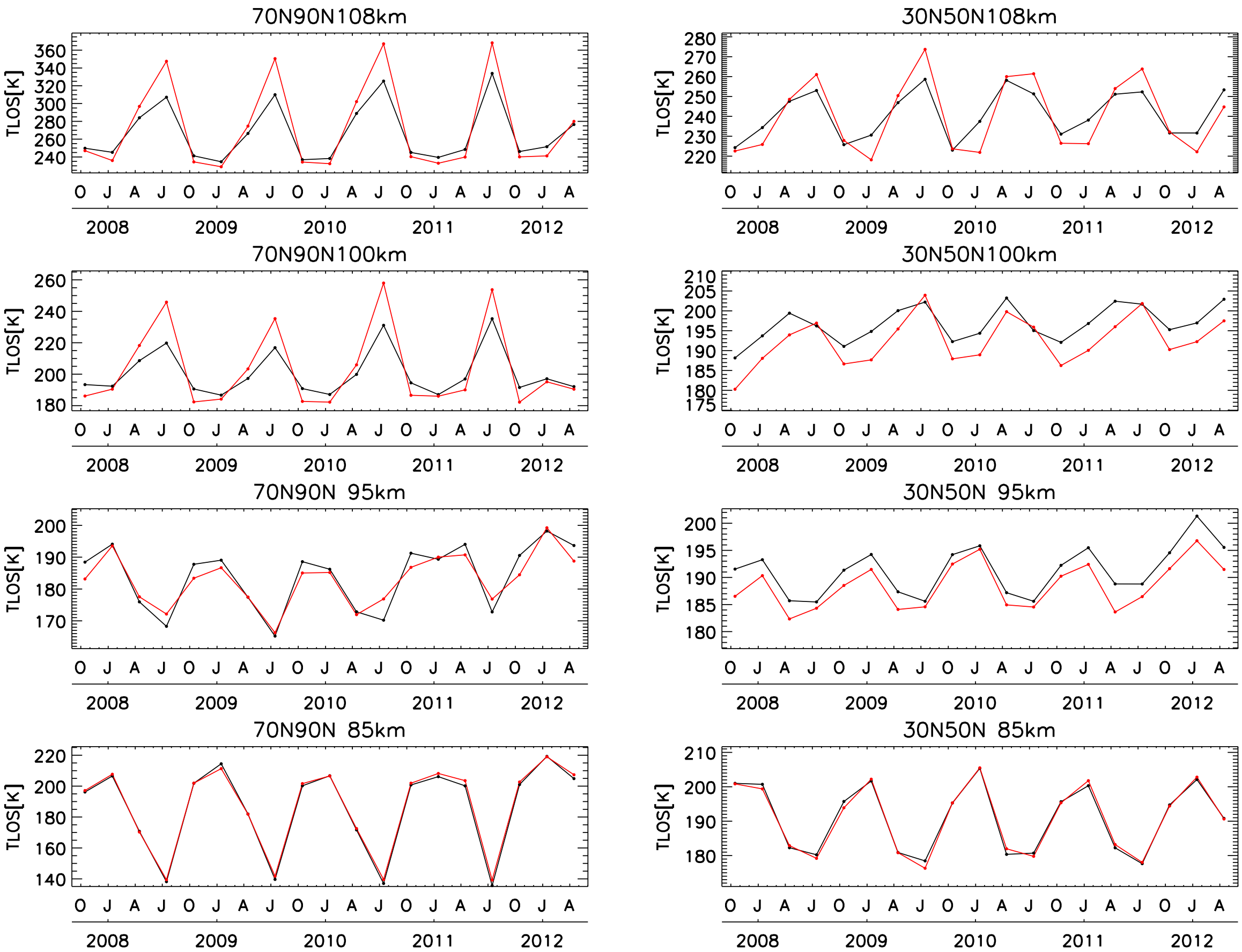

**Figure 16.** Seasonally averaged co-located MIPAS V8 (black) and SABER (red) temperatures timeseries at 108 km, 100 km, 95 km and 85 km (Left: polar latitudes (70°N–90°N); Right: mid-latitudes (30°N–50°N)).

Given its relevance for atmospheric dynamics, we also examined the MIPAS mesopause temperature and altitude time series together with those from SABER (Figs. 17 and 18). The identification of the mesopause at lower latitudes becomes difficult due to the temperature double peak structure in the upper mesosphere. Therefore, we restricted the comparison to latitudes higher than 30°.

MIPAS and SABER mesopause temperature and altitude time series at 70°–90° and 50°–70° show similar behavior for both hemispheres. MIPAS V8 high latitude winter, spring and summer mesopause temperatures are in excellent agreement with SABER (1–2 K difference), with both instruments exhibiting similar coolest temperatures during the polar summers (135–140 K in the Southern Hemisphere and, generally, 5 K colder in the Northern Hemisphere). We recall that the agreement between MIPAS V8 and SABER in spring and summer has typically improved from V5 (Fig. 12), including the year-to-year variability. This has not been the case for autumn, where the mesopause V8 co-located temperatures have continued to increase

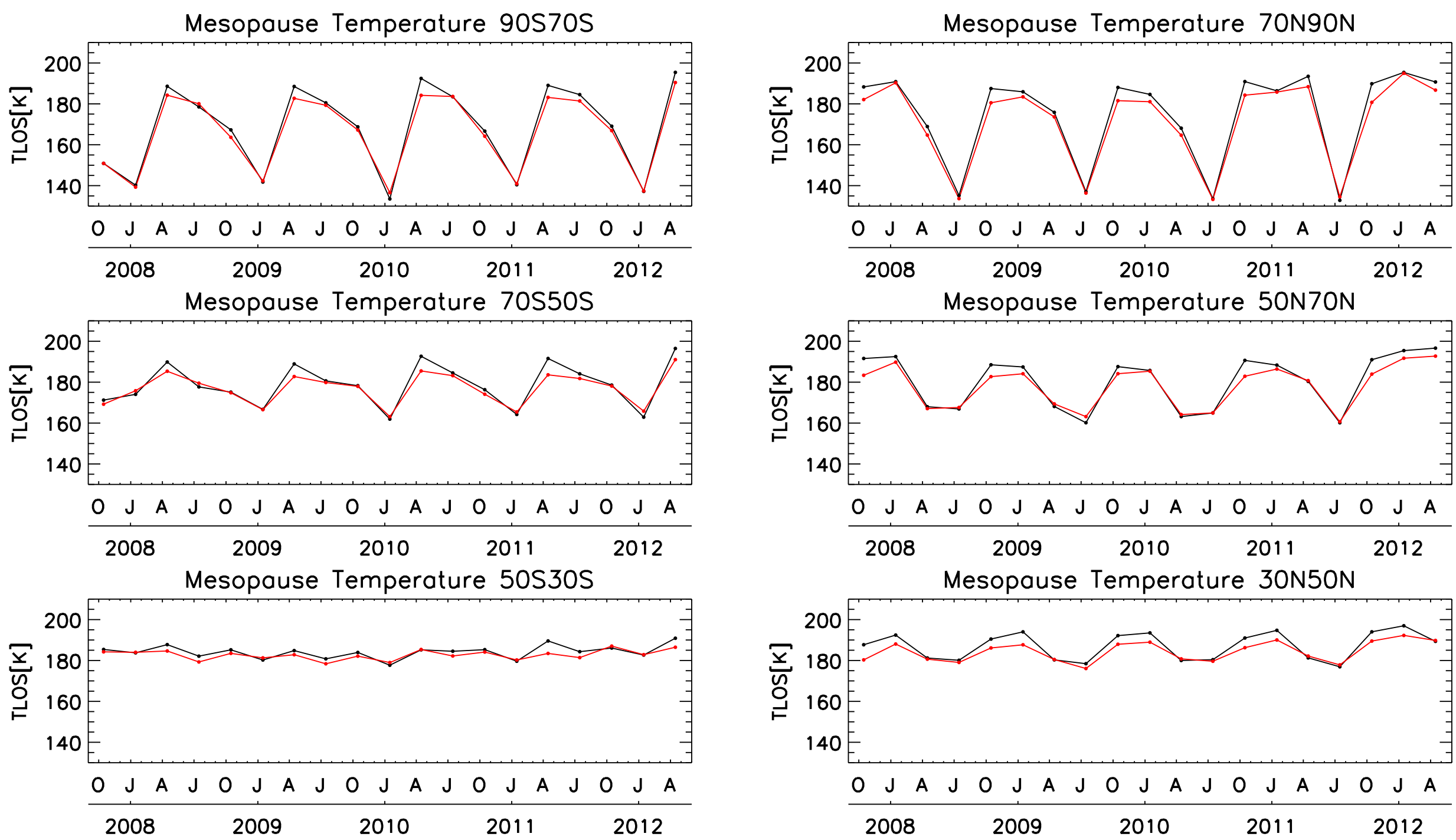

**Figure 17.** Comparison of co-located MIPAS V8 (black), MIPAS V5 (blue) and SABER (red) mesopause temperatures seasonal timeseries.

from V5, yielding a 5 K warmer autumn mesopause than SABER. This differential seasonal behaviour causes a 5 K larger autumn-to-winter mesopause temperature change and also an increased annual oscillation in V8 as compared to SABER.

Both MIPAS and SABER show very small seasonal mesopause temperature variations at 30°S–50°S (5–7 K peak-to-peak variations), barely changing from year-to-year for the time range of our comparison. The agreement is slightly worse at 30°N–50°N, where MIPAS also measures a 5 K warmer mesopause during the autumn and the winter, as for the higher latitudes.

Mesopause altitude changed differently from V8 to V5 in the Northern and Southern Hemispheres. The changes are at most 4 km during polar summers (Figs. 12–14), resulting in a generally better agreement with the SABER mesopause altitude. The comparisons of MIPAS V8 and SABER mesopause altitudes (Fig. 18) typically show a good agreement, with very similar seasonal variations. Minimum mesopause altitudes occur during summer at all latitudes, generally at 86 km in MIPAS and 88 km in SABER. The lowest co-located average mesopause altitudes measured by these two instruments were 84 km (at 50°N–70°N in 2008 and 2009 by MIPAS, and 30°N–50°N also in 2008 and 2009 by SABER). Particularly noticeable is the often parallel variation of MIPAS V8 and SABER mesopause altitudes along the time series, indicating a systematic 2 km lower mesopause in MIPAS data. An exception to this behavior occurs at 30°N–50°N, where the MIPAS autumn and summer mesopause is higher than that of SABER.

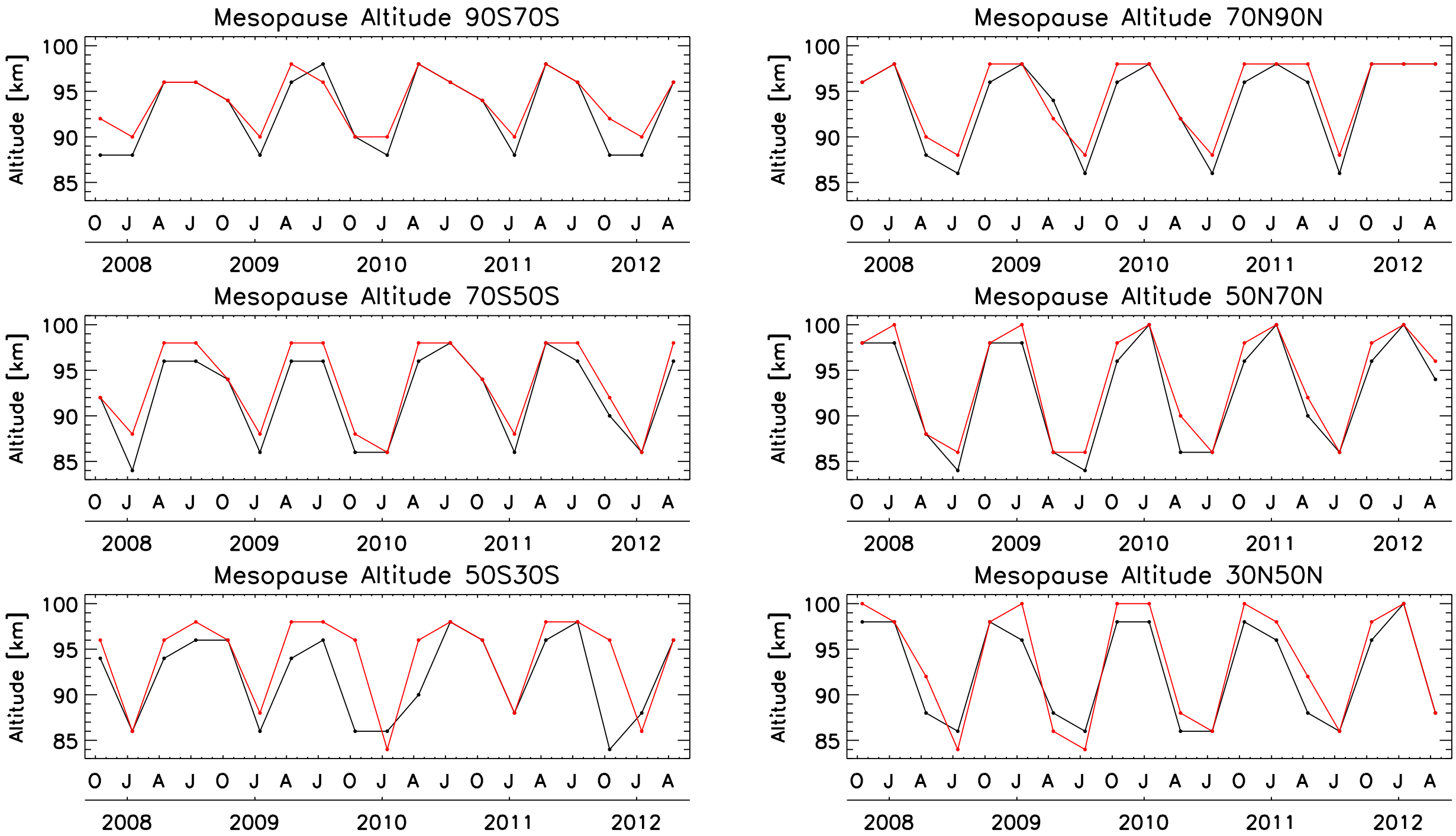

**Figure 18.** Comparison of co-located MIPAS V8 (black), MIPAS V5 (blue) and SABER (red) mesopause altitudes seasonal timeseries.

## 7 Conclusions

Following the release of version 8.03 of the MIPAS calibrated spectra by ESA, we have reprocessed the measurements in MA, UA and NLC modes to provide the community with version 8 of IMK/IAA temperature and tangent altitude correction dataset. The configuration of the IMK/IAA retrieval processor for this temperature version does not differ from that used for the inversion of the NOM data presented by Kiefer et al. (2021), except that here we took into account the non-LTE explicitly calculated for each limb scan according to the sophisticated scheme described in Funke et al. (2012), which is generally of importance above the mid-mesosphere. The use of the nearly identical retrieval configuration results in an excellent consistency between the IMK/IAA V8 NOM and MA/UA/NLC temperature products.

In addition to the improved calibrated spectra, the upgrades to the MA/UA/NLC V8 temperature retrievals with respect to the preceding IMK/IAA version 5 are: 1) more realistic atomic oxygen abundances, derived from WACCM4 abundances at MIPAS geolocations bias-corrected according to MIPAS V5 $O_3$ and O climatology; 2) more realistic carbon dioxide abundances, extracted from the SD-WACCM4 model; 3) the 2016 HITRAN spectroscopy; 4) an improved spectral shift retrieval; 5) the continuum retrieval up to 58 km; 6) the consideration of an altitude-dependent radiance offset retrieval; 7) the use of wider microwindows above 85 km in order to capture the offset; 8) a higher accuracy in the forward model calculations; 9) a new a priori temperature; 10) improved temperature horizontal gradient retrievals, where we derive a linear correction to a full 3D

temperature field; 60 km and NRLMSIS-00 temperatures above; and, 11) the use of interfering species abundances retrieved from the MIPAS IMK/IAA retrieval version 5, where available.

These upgrades in V8 result in temperature changes relative to V5 of less than 0.5 K below 70 km; 1–3 K oscillating temperature changes at low to mid-latitudes from 70 to 105 km; a vertically shifted mesopause at high-latitudes, resulting in maximum differences of 3 K in polar winter and 7 K differences in polar summer; and 5–25 K higher temperatures at 105–115 km than in the V5 version.

The MIPAS MA/UA/NLC IMK/IAA temperature dataset is reliable for scientific analysis in the full measurement vertical range for MA (18–102 km) and NLC (39–102 km), and from 42 to 115 km for UA measurements. Temperatures derived from 15 $\mu$m retrievals provided in the V8 IMK/IAA dataset do not contain atmospheric information and are not usable above 115 km.

The vertical resolution of the temperature profiles is 3 km at altitudes below 50 km, 3–5 km at 50–70 km, 4–6 km at 70–90 km, 6-1-0 km at 90–100 km and 8–11 km at 100–115 km.

We have estimated the random and systematic error components for 35 representative atmospheric conditions, covering day and night, polar, middle and tropical latitudes of the Northern and the Southern Hemispheres. The error sources considered are the measurement and the radiance offset noises, and the uncertainties in the instrument line shape, gain calibration, spectral shift, $CO_2$ spectroscopy, $CO_2$ and O abundances, and uncertainties in the collisional rates of vibrational exchange of $\nu_2$ quanta between $CO_2$ molecules, the quenching of the $CO_2(\nu_2)$ states through collisions with $N_2$ and $O_2$ and the quenching of the $CO_2(\nu_2)$ states by atomic oxygen.

The main source of MIPAS temperature random errors is the measurement noise. The noise errors are less than 1 K at altitudes below 60 km, 1–3 K at 60–70 km, 3–5 K at 70–90 km, 6–8 K at 90–100 km, 8–12 K at 100–105 km and 12–20 K at 105–115 km. Besides the measurement noise, the random component generated by the $CO_2$ abundance and the $CO_2(\nu_2)$-O quenching uncertainties are significant in the lower thermosphere, reaching 3–7 K and 5–10 K at 105–115 km.

The systematic temperature errors below 75 km are driven by uncertainties in the $CO_2$ spectroscopic data, resulting in errors smaller than 0.7 K below 55 km, 1 K at 60–70 km and smaller than 0.5 K above those altitudes. Above 80 km, the systematic temperature errors are driven by uncertainties in the non-LTE model parameters, namely the collisional rates and the atomic oxygen and $CO_2$ abundances. The non-LTE uncertainties generally lead to systematic errors smaller than 0.5 K below 75 km (80 km for the polar winter), 1 K at 80 km (2 K in the polar winter), 3 K at 95 km (5 K in the polar summer), 6–8 K at 100 km (15 K in the polar summer), 10–20 K at 105 km (30 K in the polar summer) and 20–30 K at 115 km (50 K in the polar summer). These errors are dominated by the contribution of the $CO_2(\nu_2)$-O quenching uncertainty. The $CO_2$ uncertainties lead to typical 1–2 K systematic temperature errors at 95–105 km and 7–10 K at 110–115 km. In the mid-latitude and polar summer, these values increase to 2–7 K at 95–105 km and about 20 K at 115 km.

We have conducted comparisons between MIPAS version 8 temperatures and co-located measurements from the SABER instrument. There is an excellent agreement below 90 km, with differences typically within 1.5 K, except for a few cases such as the polar summer mesosphere and the high latitude winter lower mesosphere, where differences are within 3 K. The differences between the datasets above 90 km are dependent on the season and latitude. Generally, they are within 1–3 K at 90–95 km, 1–5 K at 95–100 km, 1–8 K at 100–105 km and 1–10 K above. However, an exception to this occurs during the
polar summer above 95 km, where MIPAS and SABER differences are significantly larger (8–35 K). These differences can be roughly explained by the combined retrieval errors of both instruments, except for the polar latitudes during Autumn at 100 km and the lower mesosphere in the southern hemisphere polar winter. The comparisons between SABER and MIPAS version 8 showed an overall improvement over those with version 5, particularly at altitudes above 90 km, except for the lower thermosphere during autumn.

Our systematic error budget for temperatures highlights the need to reduce the current uncertainties in the non-LTE kinetic parameters, in particular in the $CO_2(v_2)$-O collisional rate ($k_O$), and the carbon dioxide concentrations in the upper mesosphere and lower thermosphere. Improved measurements of the rate of $CO_2(v_2)$ quenching by $N_2$ and $O_2$ ($k_{air}$) and the rate of $\nu_2$-quanta exchange between $CO_2$ molecules ($k_{vv}$) would also be beneficial. Finally, a better characterization of the instrument line shape would have also been desirable.

*Data availability.* MIPAS V8R_T_561 (MA mode), V8R_T_661 (UA mode) and V8R_T_761 (NLC mode) data are publicly accessible through https://www.imk-asf.kit.edu/english/308.php.

The supplement related to this article is available online at: https://doi.org/10.5194/amt-0-1-2023-supplement.

## Appendix A: Temperature errors values

We include in this appendix the tables with the error values corresponding to Figs. 5-7 and discussed in Sect.3. We recall that the Supplement of this manuscript contains plots and tables with the error estimations for 34 representative atmospheric scenarios, corresponding to day and night of spring, summer, autumn, and winter conditions at polar latitudes, mid-latitudes and the tropics.

*Author contributions.* MGC wrote the manuscript and had the final editorial responsibility. MGC, BF and MLP developed the retrieval setup, carried out test and verification calculations and performed the formal data analysis. UG provided and maintained the retrieval software. MK and GS took care of the inter-consistency with the nominal mode retrieval setup. NG was responsible for spectroscopy issues. SK and AL ran the retrievals. MK, NG and UG provided and maintained the error-estimation software and error estimates. TvC took care of TUNER compliance of error estimates. MGC performed the instrument inter-comparisons. MGC and BMM prepared the graphics. All the authors contributed in one way or another to the development of the retrieval methodology and its implementation, participated in the discussions, and provided text and comments.

*Competing interests.* At least one of the co-authors is a member of the editorial board of Atmospheric Measurements Techniques. The peer-review process was guided by an independent editor. The authors declare that they have no other conflict of interest.

**Table A1.** Temperature error budget for the selected solstice atmospheric scenarios at polar latitudes shown in Fig. 5. All uncertainties are 1-$\sigma$.

| Northern polar winter night | | | | | | | | | | |
|---|---|---|---|---|---|---|---|---|---|---|
| altitude (km) | mean target (K) | NLTE (K) | CO2 (K) | ils (K) | gain (K) | spectro (K) | offset (K) | noise (K) | random (K) | syst (K) |
| 20 | 208.1 | <0.1 | <0.1 | 0.2 | 0.4 | 0.4 | <0.1 | 0.3 | 0.4 | 0.6 |
| 30 | 208.7 | <0.1 | <0.1 | 0.4 | 0.4 | 0.6 | <0.1 | 0.3 | 0.6 | 0.6 |
| 40 | 225.3 | <0.1 | <0.1 | 0.1 | 0.4 | 0.5 | 0.1 | 0.4 | 0.6 | 0.6 |
| 50 | 253.9 | <0.1 | <0.1 | 0.5 | 0.4 | 0.3 | 0.1 | 0.5 | 0.6 | 0.6 |
| 60 | 246.3 | 0.2 | <0.1 | 0.3 | 0.3 | 0.8 | 0.2 | 0.9 | 1.0 | 0.9 |
| 70 | 230.2 | 0.7 | 0.3 | 0.3 | 0.2 | 0.9 | 0.8 | 2.3 | 2.5 | 1.2 |
| 80 | 218.4 | 2.1 | <0.1 | 0.5 | 0.3 | 0.7 | 1.1 | 4.4 | 4.8 | 1.7 |
| 90 | 207.3 | 2.8 | 1.8 | 1.0 | 0.5 | 0.3 | 1.4 | 5.5 | 6.0 | 2.9 |
| 100 | 195.7 | 5.8 | 1.2 | 0.9 | 0.9 | 0.3 | 1.8 | 7.6 | 8.3 | 5.3 |
| 110 | 247.2 | 19.1 | 8.5 | 1.6 | 0.7 | 0.5 | 5.6 | 21.7 | 24.4 | 18.6 |
| 115 | 301.0 | 22.7 | 11.0 | 2.2 | 0.8 | 0.6 | 7.1 | 28.3 | 31.5 | 22.5 |

| Northern polar summer day | | | | | | | | | | |
|---|---|---|---|---|---|---|---|---|---|---|
| altitude (km) | mean target (K) | NLTE (K) | CO2 (K) | ils (K) | gain (K) | spectro (K) | offset (K) | noise (K) | random (K) | syst (K) |
| 20 | 228.5 | <0.1 | <0.1 | 0.2 | 0.4 | 0.1 | <0.1 | 0.3 | 0.4 | 0.4 |
| 30 | 236.0 | <0.1 | <0.1 | 0.5 | 0.3 | 0.3 | <0.1 | 0.2 | 0.3 | 0.7 |
| 40 | 260.9 | <0.1 | <0.1 | 0.3 | 0.4 | 0.5 | <0.1 | 0.3 | 0.3 | 0.7 |
| 50 | 278.0 | <0.1 | <0.1 | 0.5 | 0.3 | 0.2 | 0.1 | 0.4 | 0.4 | 0.6 |
| 60 | 262.4 | <0.1 | <0.1 | 0.3 | 0.2 | 1.0 | 0.2 | 0.7 | 0.7 | 1.1 |
| 70 | 218.3 | <0.1 | 0.1 | 0.6 | 0.2 | 1.2 | 0.8 | 2.0 | 2.2 | 1.3 |
| 80 | 157.9 | 1.2 | 0.4 | 0.8 | 0.2 | 0.6 | 1.0 | 5.2 | 5.4 | 1.5 |
| 90 | 141.0 | 1.1 | 1.0 | 0.5 | 0.3 | 0.1 | 2.3 | 6.2 | 6.7 | 1.4 |
| 100 | 228.8 | 22.5 | 6.5 | 1.5 | 1.3 | 0.7 | 2.4 | 8.4 | 11.2 | 22.4 |
| 110 | 323.6 | 43.0 | 17.5 | 6.0 | 1.6 | 0.9 | 6.3 | 23.1 | 25.6 | 46.0 |
| 115 | 368.9 | 46.8 | 20.4 | 7.5 | 1.7 | 1.0 | 8.1 | 28.8 | 31.2 | 50.8 |

**Table A1.** (cont.)

| Southern polar summer day | | | | | | | | | | |
|---|---|---|---|---|---|---|---|---|---|---|
| altitude (km) | mean target (K) | NLTE (K) | CO2 (K) | ils (K) | gain (K) | spectro (K) | offset (K) | noise (K) | random (K) | syst (K) |
| 20 | 233.2 | <0.1 | <0.1 | <0.1 | 0.4 | 0.1 | <0.1 | 0.3 | 0.3 | 0.4 |
| 30 | 241.2 | <0.1 | <0.1 | 0.6 | 0.3 | 0.3 | <0.1 | 0.3 | 0.3 | 0.7 |
| 40 | 264.9 | <0.1 | <0.1 | 0.3 | 0.5 | 0.4 | <0.1 | 0.2 | 0.3 | 0.7 |
| 50 | 282.6 | <0.1 | <0.1 | 0.5 | 0.2 | 0.2 | 0.1 | 0.4 | 0.5 | 0.6 |
| 60 | 265.6 | <0.1 | <0.1 | 0.4 | 0.2 | 1.0 | 0.2 | 0.7 | 0.7 | 1.1 |
| 70 | 225.2 | <0.1 | 0.2 | 0.3 | 0.2 | 1.2 | 0.5 | 1.7 | 1.8 | 1.3 |
| 80 | 167.2 | 1.1 | 0.5 | 0.9 | 0.3 | 0.6 | 1.5 | 5.4 | 5.6 | 1.5 |
| 90 | 143.3 | 1.1 | 1.2 | 0.7 | 0.4 | 0.2 | 2.1 | 5.9 | 6.4 | 1.5 |
| 100 | 216.2 | 15.8 | 8.0 | 1.2 | 1.0 | 0.6 | 2.2 | 8.5 | 9.9 | 17.2 |
| 110 | 326.3 | 41.7 | 31.5 | 5.9 | 1.6 | 1.0 | 6.2 | 22.2 | 25.8 | 51.3 |
| 115 | 369.6 | 46.9 | 37.5 | 7.8 | 1.8 | 1.1 | 8.4 | 28.7 | 32.4 | 59.3 |

*Special issue statement*. This article is part of the special issue "IMK–IAA MIPAS version 8 data: retrieval, validation, and application (ACP/AMT inter-journal SI)". It is not associated with a conference.

*Acknowledgements.* The IAA team acknowledges financial support from the State Agency for Research of the Spanish MCIN through project PID2019–110689, grant CEX2021-001131-S funded by MCIN/AEI/10.13039/501100011033, and EC FEDER funds. The KIT team was supported by the Deutsches Zentrum für Luft- und Raumfahrt (DLR) under contract no. 50EE1547. The computations were partly done in the framework of a Bundesprojekt (grant MIPAS_V7) on the Cray XC40 "Hazel Hen" of the High-Performance Computing Center Stuttgart (HLRS) of the University of Stuttgart. Spectra used for this work were provided by the European Space Agency.

**Table A2.** Temperature error budget for the selected solstice atmospheric scenarios at mid-latitudes shown in the left column of Fig. 6. All uncertainties are 1-$\sigma$.

| Northern mid-latitude winter night | | | | | | | | | | |
|---|---|---|---|---|---|---|---|---|---|---|
| altitude (km) | mean target (K) | NLTE (K) | CO2 (K) | ils (K) | gain (K) | spectro (K) | offset (K) | noise (K) | random (K) | syst (K) |
| 20 | 216.8 | <0.1 | <0.1 | 0.2 | 0.4 | 0.3 | <0.1 | 0.2 | 0.4 | 0.5 |
| 30 | 221.4 | <0.1 | <0.1 | 0.5 | 0.4 | 0.4 | <0.1 | 0.3 | 0.4 | 0.7 |
| 40 | 240.2 | <0.1 | <0.1 | 0.2 | 0.4 | 0.6 | <0.1 | 0.3 | 0.4 | 0.7 |
| 50 | 251.1 | <0.1 | <0.1 | 0.5 | 0.3 | 0.5 | 0.1 | 0.5 | 0.6 | 0.7 |
| 60 | 234.4 | 0.1 | <0.1 | 0.4 | 0.3 | 0.9 | 0.2 | 0.9 | 1.0 | 1.0 |
| 70 | 221.3 | 0.4 | 0.3 | 0.5 | 0.2 | 0.8 | 0.7 | 2.3 | 2.4 | 1.0 |
| 80 | 215.1 | 1.7 | <0.1 | 0.5 | 0.3 | 0.7 | 1.1 | 4.4 | 4.7 | 1.6 |
| 90 | 197.3 | 2.1 | 1.5 | 1.0 | 0.4 | 0.3 | 1.5 | 5.7 | 6.1 | 2.4 |
| 100 | 195.4 | 6.9 | 1.3 | 0.5 | 0.9 | 0.5 | 1.8 | 7.4 | 8.1 | 6.5 |
| 110 | 241.6 | 19.9 | 8.3 | 1.6 | 0.6 | 0.5 | 5.4 | 21.0 | 24.5 | 18.3 |
| 115 | 304.3 | 24.5 | 11.1 | 2.5 | 0.6 | 0.6 | 7.0 | 27.9 | 32.1 | 23.1 |

| Northern mid-latitude summer day | | | | | | | | | | |
|---|---|---|---|---|---|---|---|---|---|---|
| altitude (km) | mean target (K) | NLTE (K) | CO2 (K) | ils (K) | gain (K) | spectro (K) | offset (K) | noise (K) | random (K) | syst (K) |
| 20 | 221.5 | <0.1 | <0.1 | 0.2 | 0.4 | 0.1 | <0.1 | 0.3 | 0.4 | 0.4 |
| 30 | 232.7 | <0.1 | <0.1 | 0.5 | 0.4 | 0.4 | <0.1 | 0.3 | 0.3 | 0.7 |
| 40 | 256.0 | <0.1 | <0.1 | 0.3 | 0.5 | 0.5 | <0.1 | 0.3 | 0.3 | 0.7 |
| 50 | 267.0 | <0.1 | <0.1 | 0.5 | 0.3 | 0.3 | 0.1 | 0.4 | 0.5 | 0.7 |
| 60 | 246.1 | <0.1 | <0.1 | 0.3 | 0.2 | 1.1 | 0.3 | 0.8 | 0.8 | 1.2 |
| 70 | 204.8 | 0.2 | 0.1 | 0.6 | 0.2 | 0.9 | 0.8 | 2.4 | 2.5 | 1.1 |
| 80 | 165.0 | 1.1 | 0.2 | 0.5 | 0.2 | 0.4 | 1.0 | 5.1 | 5.2 | 1.1 |
| 90 | 169.3 | 1.0 | 1.1 | 1.3 | 0.6 | 0.2 | 1.8 | 6.1 | 6.5 | 1.8 |
| 100 | 210.3 | 14.0 | 3.8 | 0.5 | 0.8 | 0.4 | 2.0 | 7.6 | 8.9 | 14.0 |
| 110 | 311.4 | 39.1 | 16.9 | 4.0 | 1.2 | 0.8 | 5.6 | 21.9 | 25.5 | 41.1 |
| 115 | 374.8 | 43.7 | 20.2 | 5.4 | 1.3 | 0.9 | 7.5 | 27.8 | 31.4 | 46.8 |

**Table A3.** Temperature error budget for selected equinox atmospheric scenarios at mid-latitudes shown in the right column of Fig. 6. All uncertainties are $1\sigma$.

| Southern mid-latitude spring day | | | | | | | | | | |
|---|---|---|---|---|---|---|---|---|---|---|
| altitude (km) | mean target (K) | NLTE (K) | CO2 (K) | ils (K) | gain (K) | spectro (K) | offset (K) | noise (K) | random (K) | syst (K) |
| 20 | 223.1 | <0.1 | <0.1 | 0.1 | 0.4 | 0.2 | <0.1 | 0.2 | 0.4 | 0.4 |
| 30 | 229.7 | <0.1 | <0.1 | 0.4 | 0.4 | 0.3 | <0.1 | 0.3 | 0.4 | 0.6 |
| 40 | 246.1 | <0.1 | <0.1 | 0.3 | 0.4 | 0.5 | <0.1 | 0.3 | 0.4 | 0.7 |
| 50 | 261.4 | <0.1 | <0.1 | 0.5 | 0.3 | 0.4 | 0.1 | 0.4 | 0.5 | 0.7 |
| 60 | 243.3 | <0.1 | <0.1 | 0.3 | 0.2 | 1.1 | 0.2 | 0.8 | 0.8 | 1.1 |
| 70 | 212.7 | 0.1 | 0.2 | 0.5 | 0.2 | 0.9 | 0.7 | 2.3 | 2.4 | 1.0 |
| 80 | 193.9 | 0.6 | 0.1 | 0.4 | 0.2 | 0.6 | 1.4 | 5.0 | 5.2 | 0.7 |
| 90 | 183.9 | 1.3 | 1.4 | 0.9 | 0.4 | 0.2 | 1.4 | 5.7 | 6.0 | 1.7 |
| 100 | 188.4 | 8.5 | 2.2 | 0.6 | 1.0 | 0.6 | 2.2 | 8.4 | 9.3 | 8.3 |
| 110 | 271.4 | 27.0 | 10.0 | 3.3 | 1.1 | 0.7 | 5.7 | 21.5 | 23.7 | 27.8 |
| 115 | 341.9 | 32.7 | 13.4 | 4.7 | 1.2 | 0.9 | 7.5 | 28.3 | 30.8 | 34.3 |

| Northern midlatitude autumn night | | | | | | | | | | |
|---|---|---|---|---|---|---|---|---|---|---|
| altitude (km) | mean target (K) | NLTE (K) | CO2 (K) | ils (K) | gain (K) | spectro (K) | offset (K) | noise (K) | random (K) | syst (K) |
| 20 | 216.5 | <0.1 | <0.1 | 0.2 | 0.4 | 0.2 | <0.1 | 0.3 | 0.4 | 0.4 |
| 30 | 222.8 | <0.1 | <0.1 | 0.4 | 0.4 | 0.3 | <0.1 | 0.3 | 0.3 | 0.6 |
| 40 | 241.3 | <0.1 | <0.1 | 0.2 | 0.4 | 0.5 | <0.1 | 0.3 | 0.4 | 0.7 |
| 50 | 253.4 | <0.1 | <0.1 | 0.5 | 0.3 | 0.4 | 0.1 | 0.4 | 0.5 | 0.7 |
| 60 | 237.3 | <0.1 | <0.1 | 0.3 | 0.2 | 1.0 | 0.3 | 0.9 | 0.9 | 1.0 |
| 70 | 216.3 | 0.3 | 0.2 | 0.5 | 0.2 | 1.0 | 0.8 | 2.4 | 2.6 | 1.1 |
| 80 | 196.7 | 0.8 | <0.1 | 0.6 | 0.2 | 0.7 | 1.0 | 4.8 | 4.9 | 0.9 |
| 90 | 197.0 | 2.5 | 1.4 | 1.1 | 0.4 | 0.2 | 1.4 | 5.8 | 6.1 | 2.6 |
| 100 | 191.2 | 6.7 | 1.0 | 0.6 | 1.0 | 0.5 | 1.8 | 7.5 | 8.1 | 6.5 |
| 110 | 231.5 | 18.8 | 4.8 | 1.6 | 0.7 | 0.4 | 4.7 | 19.1 | 21.5 | 17.5 |
| 115 | 286.9 | 23.1 | 6.8 | 2.5 | 0.8 | 0.5 | 6.3 | 27.0 | 29.7 | 21.7 |

**Table A4.** Temperature error budget for Tropics day, also shown in Fig. 7. All uncertainties are 1-$\sigma$.

| Tropics day | | | | | | | | | | |
|---|---|---|---|---|---|---|---|---|---|---|
| altitude (km) | mean target (K) | NLTE (K) | CO2 (K) | ils (K) | gain (K) | spectro (K) | offset (K) | noise (K) | random (K) | syst (K) |
| 20 | 203.2 | <0.1 | <0.1 | <0.1 | 0.2 | 0.8 | <0.1 | 0.2 | 0.3 | 0.8 |
| 30 | 230.1 | <0.1 | <0.1 | 0.5 | 0.3 | 0.5 | <0.1 | 0.3 | 0.3 | 0.7 |
| 40 | 254.4 | <0.1 | <0.1 | 0.3 | 0.4 | 0.6 | <0.1 | 0.3 | 0.4 | 0.7 |
| 50 | 263.7 | <0.1 | <0.1 | 0.5 | 0.3 | 0.4 | 0.1 | 0.4 | 0.5 | 0.7 |
| 60 | 241.9 | <0.1 | <0.1 | 0.4 | 0.2 | 1.1 | 0.3 | 0.8 | 0.9 | 1.2 |
| 70 | 207.7 | 0.2 | 0.2 | 0.6 | 0.2 | 1.0 | 0.7 | 2.3 | 2.5 | 1.1 |
| 80 | 190.0 | 0.6 | 0.1 | 0.7 | 0.2 | 0.5 | 1.1 | 4.8 | 4.9 | 0.8 |
| 90 | 193.4 | 1.6 | 1.5 | 1.1 | 0.4 | 0.2 | 1.4 | 5.7 | 6.1 | 2.0 |
| 100 | 190.8 | 5.7 | 1.3 | 0.5 | 0.7 | 0.3 | 1.8 | 7.0 | 7.6 | 5.4 |
| 110 | 239.9 | 19.1 | 7.6 | 1.9 | 0.8 | 0.4 | 4.5 | 18.7 | 21.3 | 18.6 |
| 115 | 295.7 | 23.8 | 10.4 | 3.2 | 0.9 | 0.6 | 6.1 | 26.5 | 29.5 | 23.6 |

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
