# Peer review of "Version 8 IMK/IAA MIPAS temperatures from 12–15 $\mu$ m spectra: Middle and Upper Atmosphere modes"

_Atmospheric Measurement Techniques, 2023_

## Author Comment (AC1)

**Referee #2**

*Reply to referee #2 for the review of the manuscript*:
**Version 8 IMK/IAA MIPAS temperatures from 12–15 µm spectra: Middle and Upper Atmosphere modes (amt-2023-119)**

We thank the reviewer for his/her valuable comments and suggestions. We have addressed all of them all. In this response, we go through the raised issues point by point and outline the changes we intend to make (see attached file). Additionally, we have corrected several typos.

**Reviewer's summary:**
*The paper reports the new retrieval of temperature in the mesosphere using MIPAS latest realeased level 1b spectra. I am not a native English speaker, therefore I just made few language corrections. It deserves to be published after the following comments have been addressed.*

**Reviewer's suggestion**:
*Line 29 : Add some reference to the ESA site or to the official document*

**Author's response**:
We will add a reference to the corresponding ESA file:
ESA: Product Quality Readme File for MIPAS Level 1b IPF 8.03 products – issue 1.1, ESA-EOPG-EBA-TN-1, available at:
https://earth.esa.int/eogateway/documents/20142/37627/Read_Me_File_MIP_NL__1PY_ESA-EOPG-EBA-TN-1-issue1.1.pdf (last access: September 2023), 2019.

**Reviewer's suggestion**:
*Line 34: This part is officially known as Optimised Resolution, please add this name when you first mention the part of the measurements*

**Author's response**:
We will add to the text: 'also called optimised resolution by ESA'.

**Reviewer's suggestion**:
*Line 39: substitute 'a few days in a row' with 'few (or write the number if it always the same) consecutive days'*

**Author's response**:
We will replace it by 'few consecutive days'.

**Reviewer's suggestion**:
*Line 70: I assume each dataset refers to a single observation mode, I suggest making it clearer in the text here.*

**Author's response**:
That is correct. We will add 'respectively' after 'dataset'.

**Reviewer's suggestion**:
*Line 114-117: please add some reference to documents or papers that describe this.*

**Author's response**:
We will add the corresponding references.

**Reviewer's suggestion**:
*Line 124-125: there is a new release of HITRAN (2020) why did you not use it?*

**Author's response**:
Our processing of the MIPAS v8 data began in 2019, prior to the release of HITRAN2020. In that processing, not only temperature but other numerous atmospheric species were retrieved, for which this version 8 of temperatures were used. For consistency, the HITRAN2016 version has been maintained in all retrievals.

**Reviewer's suggestion**:
*Line 137: But you do not retrieve continuum above 58 km, so why you find problems?*

**Author's response**:
The reviewer is right to ask this question. In fact, the altitude was not correct: the heights where the offset correction cannot be distinguished are also below 60 km (in particular, in the regions where radiative transfer is linear). This sentence will be changed in the revised version, replacing the value of 60 km by ~30 km.

**Reviewer's suggestion**:
*Line 144: substitute 'km in just' with 'km just'*

**Author's response**:
We will do it in the revised version .

**Reviewer's suggestion**:
*Line 154-155: The sentence should be changed in 'In V5R_t_m21, the temperature ….etc.'*

**Author's response**:
We agree.

**Reviewer's suggestion**:
*Line 192: Substitute 'in each iteration' with 'at each iteration'*

**Author's response**:
We will change it.

**Reviewer's suggestion**:
*Line 193: Substitute 'change slightly' with 'have been slightly changed'*

**Author's response**:
We will do it in the revised version.

**Reviewer's suggestion**:
*Line 200: Substitute 'is' with 'are'*

**Author's response**:
We will change it. Thank you.

**Reviewer's suggestion**:
*Line 208-210: Hard to see what you say in the figure due to the adopted color scale. Also checking the AK it looks that very small information is contained in the measurements above 100 km, so this comment doesn't hold very well*

**Author's response**:
We agree that the color scale does not help. However, we think that the overplotted contour lines do. This is particularly true for the plots for solstice, where the isolines are inclined down towards the polar summer.
We also think there is considerable information up to 110 km in MA and NLC (averaging kernel diagonal ~0.18) and up to 115 km in UA (averaging kernel diagonal ~0.26), as the averaging kernels in Fig. 2 show.

**Reviewer's suggestion**:
*Lines 213-215: Again very hard to see in the figure*

**Author's response**:
This refers to the yellowish blob around 85 km at low latitudes during equinox. We will mention this explicitly in the text to make it easier to find.

**Reviewer's suggestion**:
*Lines 219-222: Could you please explain better this point, I could not follow it! What 'elevated stratopause events' are?*

**Author's response**:
Elevated stratopauses were mentioned in Labitzke (1972), described in detail in Manney et al. (2005). The term 'Elevated stratopause' refer to extreme events in which the stratopause reforms at uncommonly high altitudes (in the middle-upper mesosphere), occasionally occuring after sudden stratospheric warmings ( see e.g., Manney et al. (2005), Siskind et al., 2007; Chandran et al., 2011, 2013; McLandress et al., 2013; Limpasuvan et al., 2016). We will add the reference to Manney et al. (2005) in the revised version:
Manney, G. L., Krüger, K., Sabutis, J. L., Sena, S. A., and Pawson, S.: The remarkable 2003–2004 winter and other recent warm winters in the Arctic stratosphere since the late 1990s, J. Geophys. Res., 110, D04107, https://doi.org/10.1029/2004JD005367, 2005,

**Reviewer's suggestion**:
*Line 224: 'wrinkled' is not very scientific, use oscillating or something more appropriate.*

**Author's response**:
We chose "wrinkled" because we wanted to emphasize that it is an irregular structure. We would use "oscillating" to denote a regular structure. With this in mind and if the reviewer agrees, we prefer to stick to "wrinkled" in this context.

**Reviewer's suggestion**:
*Line 230: what is the threshold you use of the AK peak to decide if there is information?*

**Author's response**:
We should have written the threshold in the text, as the reviewer suggests. We will add the following sentence:

"Our recommendation for the data users is to discard data points at altitudes where the corresponding averaging kernel diagonal element is less than 0.03, because they lack significant measurement information."

**Reviewer's suggestion**:
*Lines 235-239: Could you explain why the vertical resolution oscillates so much?*

**Author's response**:
This is a natural phenomenon characteristic of the retrieval when using more retrieved points (state vector) and measurements. and there is nothing wrong with it: at a tangent altitude (where the instrument provides measurements), the vertical resolution is better but the error by measurement noise is larger. Between two tangent altitudes, where the results depend on information of two measurements (above and below), noise is smaller but the resolution is worse. If averaging kernels were
evaluated only on the grid given by the tangent altitudes, this effect would not be visible.
We will include a sentence in the revised manuscript clarifying this behavior:
"The oscillating behavior depicted arises because the vertical resolution is better at retrieval altitudes closer to the tangent altitudes (see Fig. 4), despite the larger error introduced by measurement noise."

**Reviewer's suggestion**:
*Line 245: you say 34 scenarios: shouldn't them be 40? 5 latitude bands X 4 seasons X 2 (day/night)*

**Author's response**:
Please, note that we do not distinguish between seasons for tropical latitudes.

**Reviewer's suggestion**:
*Line 249: You say 'in the following', is it in the table or in the text?*

**Author's response**:
The expression was certainly not clear. In the revised version, we will replace: "In the following, we provide 1-sigma uncertainties." with "All reported uncertainties are one sigma throughout."

**Reviewer's suggestion**:
*Line 253: 'propagation of measurement' -> 'propagation of the measurement'*

**Author's response**:
We agree. We will change the text accordingly.

**Reviewer's suggestion**:
*Line 255: The noise values change very much from the region around 700 cm-1 to the region at 900 cm-1, I would rather use the range than the average value here*

**Author's response**:
We will provide the noise range in the revised version. We would like to note that we do not use the entire band up to 900 cm-1.

**Reviewer's suggestion**:
*Lines 256-259: The calibration is performed as part of your analysis or as part of the level1 process?*

**Author's response**:
In the revised version, we will specify that: "(…) there is still a remaining random uncertainty due to the wavelength dependence of the deep space measurements used for the level-1b radiance offset calibration."

**Reviewer's suggestion**:
*Lines 260-261: do you analyse unapodised spectra? Otherwise the ILS shape should be dominated by the apodization used*

**Author's response**:
We use apodized spectra, and this apodization is considered in the error estimation. It still does make an effect, particularly at higher altitudes.

**Reviewer's suggestion**:
*Table 3: even if you explain it in the caption the word chief does not suggest what you have introduced in the column, could you use a different and more appropriate word?*

**Author's response**:
We will substitute 'chief' by 'Char.' (from characteristic) in the column title.

**Reviewer's suggestion**:
*Lines 275-278: HITRAN usually reports spectroscopic uncertainties, why haven't you used them?*

**Author's response**:
Unfortunately, HITRAN spectroscopic uncertainties are not completely helpful because it is not reported whether they are 1-sigma, 2-sigma or 3-sigma. Contacting the HITRAN lead author did not help either. Thus, we decided to contact an expert in the field to provide these uncertainties (Manfred Birk, February 2020), as it is stated in the text.

**Reviewer's suggestion**:
*Lines 279-283: the whole paragraph is rather confusing, could you explain it better? An educated guess is not a quantity, could you explicit the guess you made?*

**Author's response**:
We will try to be clearer in the revised version. We shall specify the uncertainty used for the calculations. We will also re-write the paragraph:
"For gases derived in preceding MIPAS V5 retrievals, we have used the noise covariance information obtained from those retrievals. For interfering gases not previously retrieved from MIPAS measurements, we relied on our initial guess database (Kiefer et al., 2002, and updates thereof). In the latter case, accurate uncertainty information for these abundances is often unavailable and we have assumed a 100\% uncertainty."

**Reviewer's suggestion**:
*Lines 296-297: Some of the random errors are systematic, but of a random nature….*

**Author's response**:
We are not certain if we have interpreted this comment correctly. In the context of data analysis and, consequently, for the benefit of data users, we believe it is advantageous to provide systematic and random errors separately. For instance, systematic errors will result in biases that need not be considered when analyzing variations relative to mean values. This is

discussed in Clarmann et al. (2020), as referenced in the text, and is beyond the scope of this manuscript.

**Reviewer's suggestion**:
*Lines 305-306: Do you mean that you did not know the covariance matrix associated to those errors?*

**Author's response**:
It depends on the error source. For instance, there was no covariance information available for $CO_2$. However, for most of the other error sources that we treated using the perturbation method, the covariance treatment is unnecessary since the uncertainties pertain to single, independent parameters. In other words, there would be little gained by using the covariance formalism because we are only dealing with the variances of the single, independent parameters.
In order to be more precise, we will add "due to unavailability of covariance information" after "not possible".

**Reviewer's suggestion**:
*Lines 307-309: The whole sentence is not very clear, could you describe this better?*

**Author's response**:
We propose to re-write the sentence as follows:
"There were error sources for which no specific values of their random uncertainty could be prescribed (for instance, $CO_2$ abundance and spectroscopy, non-LTE, ILS). Since temperature errors triggered by a systematic parameter uncertainty also exhibit a random component due to atmospheric variability, we also conducted statistically analyses on the responses to perturbations across the ensemble of temperature profiles, each ensemble representing a distinct atmospheric scenario. We took the dispersion within these responses as our estimate of the associated temperature error random component for those cases."

**Reviewer's suggestion**:
*Line 313: what do you mean with 'chiefly'?*

**Author's response**:
We mean 'not completely, but as a most important part', a synonym of 'mainly' or 'primarily'. Although we think both "Chiefly random errors" and "chiefly systematic errors" are grammatically and semantically correct, we will change word order in this sentence to write "Errors chiefly random" and "errors chiefly systematic", in the hope that the expression can be better understood.

**Reviewer's suggestion**:
*Line 315 'for 34' -> 'for the 34'*

**Author's response**:
We agree.

**Reviewer's suggestion**:
*Line 430: Which NLTE model does Saber use? If it is the same as in this paper it should be mentioned, as in this altitude range it is the main contributor to the retrieved temperatures.*

**Author's response**:

The references where the SABER non-LTE model is described are provided in the manuscript (Remsberg et al., 2008; Garcia-Comas et al. 2008). The SABER non-LTE algorithm is not the same as the MIPAS non-LTE model. However, some non-LTE parameters are set equal. That is the case for the quenching rates of $CO_2(v2)$ by O, N2 and O2, but, for instance, not the case for atomic oxygen abundance or the rate of CO2-CO2 v2-quanta exchange. Consequently, for our SABER and MIPAS comparisons, we removed the contribution from the CO2-O, CO2-N2 and CO2-O2 quenching rate uncertainties from the combined errors and the discussion in the text takes this fact into account. This is specifically stated in lines 447-450.

**Reviewer's suggestion**:
*Line 481 onward: I did understand that the comparison were done on the smoothed profiles only, so please clarify this better in the text!*

**Author's response**:
We propose to change the wording:
"Beyond this, we have also compared collocated MIPAS and SABER time series. For this purpose, MIPAS averaging kernels and a priori information were applied to the SABER profiles."

**Reviewer's suggestion**:
*In the references at  Line 699 ' Fera, S.D.' -> ' Della Fera, S.'*

**Author's response**:
Thank you. We will change this name in the reference.

---

## Author Comment (AC2)

**Referee #1**

*Reply to referee #1 for the comment on the manuscript* **amt-2023-119:**
**Version 8 IMK/IAA MIPAS temperatures from 12–15 µm spectra: Middle and Upper Atmosphere modes**

We thank the reviewer for his/her valuable comments and suggestions. We have addressed all of them. In this response, we go through the raised issues point by point and outline the changes we intend to make. Additionally, we have corrected several typos.

*Review of "Version 8 IMK/IAA MIPAS temperatures from 12-15 micron spectra: Middle and Upper Atmosphere modes" by Garcia-Comas et al.*

*General Comments:*

**Reviewer's summary:**
*This manuscript describes a new MIPAS temperature dataset from 18-115 km altitude, which the authors call version 8.03. In addition to improved calibrated spectra, the authors detail a variety of improvements to the retrieval and show several comparisons against previous versions of the data as well as comparisons against existing SABER temperature observations. The paper provides a comprehensive description of the uncertainties and overall, the figures are instructive and clear. Although the authors spend considerable effort comparing their results to SABER temperatures, comparisons with recently re-processed MIPAS temperatures using nitric oxide emission are quite limited and it is not really clear why. The manuscript is also lacking detail in some places, particularly in explaining the results in the upper altitude region of their dataset. The reviewer recommends publication provided that the authors address the concerns enumerated below.*

*Specific Comments:*

**Reviewer's suggestion**:
*Line 97 and elsewhere. The authors spend considerable effort comparing to existing SABER temperature observations. Could the authors please comment on and show how their results compare to MIPAS thermospheric temperatures derived from nitric oxide spectra at 5.3 microns (e.g. Funke et al., Atmos. Meas. Tech., 16, 2167-2196, 2023)? There is mention of day-night differences on lines 384-387 and latitude variations on lines 394-396. However, a direct comparison between the two MIPAS temperature datasets would be particularly useful for the reader. If there is a reason for this limited comparison or if this has been done in other publications then please say so and give a reference. Thank you.*

**Author's response**:
We think this is a very good suggestion.
We propose to rename Section 5 with "Consistency with NOM mode 12-15-micron and UA mode 5.3-micron temperatures" and to include the following paragraph at the end of Section 5:
"Version 8 temperature retrievals from nitric oxide spectra at 5.3 microns, obtained by MIPAS in the UA mode, are detailed in Funke et al. (2023). The a priori temperature profiles used to derive those 5.3-micron temperatures consist solely of version 8 15-micron temperatures below 110 km (presented here) and solely of corrected MSIS version 2.0 temperature profiles above 120 km (Emmert et al., 2021). A merging of the two is performed at 110-120 km, as

described in Funke et al. (2023). Moreover, retrieved temperatures are strongly regularized towards 12-15-micron our temperatures below 110 km.

We examined the differences between our 12-15-micron temperatures and version 8 5.3-micron temperatures (not shown). Below 110 km, both temperature datasets are essentially identical for all atmospheric conditions, as expected, except for summer at latitudes poleward of 30º, where the differences are within 5K. At 115 km, the 5.3-micron temperatures are generally 5-10K smaller than our 12-15-micron temperatures, and this difference can reach 20-30K for summer at latitude poleward of 30º. Importantly, the differences do not generally depend on solar illumination. Therefore, the differences of the datasets are within our estimated systematic errors for all atmospheric scenarios."

**Reviewer's suggestion**:

*Lines 165-179. Could the authors show how the model results used for atomic oxygen compare to the recent MSIS 2.0 for atomic oxygen (Emmert et al., Earth and Space Science, 7, e2020EA001321. https://doi.org/10.1029/2020EA001321, 2020)? More importantly, how do any differences in MSIS 2.0 atomic oxygen affect the temperature retrieval?*

**Author's response**:

We thank the reviewer for this comment, that let us find a mistake in the text (Line 293). Section 3 of the manuscript describes the the temperature error sources, including those coming from the uncertainties in the atomic oxygen abundances used in our retrievals. The estimation of the atomic oxygen uncertainty above 95 km has actually been done by means of comparisons with NRLMSIS 2.0 data, and not with NRMLMSISE00 data, as it was written in the text (lines 289-295). Therefore, Fig. 4 actually shows the average uncertainty based on comparisons with MSIS 2.0 atomic oxygen above 95km merged to the estimated MIPAS V5 atomic oxygen error below 95km (note that these are daytime and nighttime averages of the distinct abundances that we took for five latitude boxes and for the four seasons). Moreover, the effect of the differences between the atomic oxygen used and that of MSIS 2.0 on the retrieved temperatures is therefore included implicitly in the O-uncertainty component of our temperature non-LTE error estimation, which is described in Section 3.2 (see blue lines in Fig. 8). We would like to emphasize that, firstly, the effect of atomic oxygen uncertainty on retrieved temperature is only important above 95 km (see Fig. 8) and, secondly, the estimated MIPAS V5 atomic oxygen uncertainty is generally larger than that based on comparisons with MSIS 2.0 below 95 km.

We will correct the MSIS model version number in the text and will include the corresponding reference (Emmert et al., 2020) in the revised version.

**Reviewer's suggestion**:

*Line 230 and Figure 2. Are these kernels available to the reader? They would be useful for comparing with models or other datasets, but it is not clear from the data availability statement at the end of the paper that they are available. Note also that the reviewer could not connect using the link to the supplement at the end of the paper.*

**Author's response**:

Our data server does not regularly provide the individual averaging kernels for each temperature profile, due to the excessive volume of these data. Nevertheless, they can be provided upon request. In the data availability statement of the revised version, we will therefore state:" Due to their data volume, averaging kernels can be made available upon request."

The link to the supplement will be updated.

**Reviewer's suggestion**:
*There are many acronyms throughout this abstract that are not spelled out. Please correct these when first mentioned. Thank you.*

**Author's response**:
We will try to find all acronyms and spell them out when first mentiones in the revised manuscript. We note here that we will replace "V8" and "V5" with "version 8" and "version 5".

**Reviewer's suggestion**:
*Lines 25-26. Please give a number here (+/- X K) and indicate the altitudes at which this applies. SABER faces similar non-LTE challenges in that temperature retrieval. From Figures 12-15 the agreement gets worse and the combined systematic uncertainties get much larger above 90 km so the authors need to be more explicit about this in the abstract.*

**Author's response**:
We will replace the last sentence in the abstract by the following sentences:
"The comparison of this V8 temperature dataset with co-located SABER temperature measurements shows an excellent agreement, with differences typically within 1.5 K  below 90 km, 1–3 K at 90–95 km, 1–5\,K at 95–100 km, 1–8 K at 100–105 km and 1–10 K above, that is, all falling within the combined errors. The agreement with SABER improves with respect to previous MIPAS IMK/IAA versions."

**Reviewer's suggestion**:
*Line 30. Please give a reference for these v5 spectra.*

**Author's response**:
We will include the reference in the revised version.

**Reviewer's suggestion**:
*Line 80. A brief paragraph describing the Envisat mission, as well as the orbital inclination and the equator crossing local times would be very helpful to the uninitiated reader here in order to give valuable context to the dataset.*

**Author's response**:
We will include the following sentences at the beginning of the second paragraph in Section 1:
"The ENVISAT satellite was placed into a polar Sun-synchronous orbit, with an inclination of 81.5º and an altitude of approximately 800 km. The orbital period was about 101 minutes, enabling the satellite to complete a global Earth coverage in 14.3 daily orbits. The Equator crossing local times were approximately 10:00 and 22:00."

**Reviewer's suggestion**:
*Line 149. What do the authors mean by "bias-corrected"? Is this a bias in the model and if so, why is this done? Given the importance of atomic oxygen in the temperature retrievals, one or two sentences explaining this would be helpful.*

**Author's response**:
We are not sure if the Line number indicated by the Reviewer is correct because that line describes the a priori temperature. Since the reviewer explicitly mentions atomic oxygen in

this question, we think he refers to Line 167, where the bias-correction related to atomic oxygen is discussed.

The description of the "bias-correction" performed to the WACCM4 Ox to derive the atomic oxygen used in the temperature retrievals is actually written in the three paragraphs in Lines 168-179. To emphasize these all, describe the bias-correction, we have combined the three paragraphs into one.

In order to provide a justification for this correction, we will include the following sentence: "This approach is made to preserve the transient variability as provided by WACCM4 while retaining the climatological atomic oxygen as derived from MIPAS measurements in the previous version".

**Reviewer's suggestion**:
*Figure 2 caption. To what does 30 degrees refer?*

**Author's response**:
The words "latitude of" before the number were missing.

**Reviewer's suggestion**:
*Line 270. The reviewer does not understand how CO2 uncertainties can be calculated from the WACCM results. Why can this uncertainty not be estimated by ACE and/or SABER data alone? Please explain in the text.*

**Author's response**:
The uncertainties are estimated from WACCM CO2 (at MIPAS geolocations) by means of comparisons with ACE and SABER observations because that is the WACCM CO2 abundance that we use as input for these temperature retrievals, as written in Line 163. Nevertheless, for clarity, we have added "at MIPAS geolocations" after WACCM CO2 in this sentence.

**Reviewer's suggestion**:
*Lines 289-295. Again, the reviewer does not understand how an atomic oxygen uncertainty can be calculated from the WACCM results. Please explain in the text. Furthermore, it is well known that the NRLMSIS00 atomic oxygen values above 90 km are inaccurate (e.g. Sheese et al., J. Geophys. Res., 116, D01303, doi:10.1029/2010JD014640). The reviewer requests that the authors either crop their Fig. 4 at 97 km or use the newer more accurate values from MSIS 2.0 (Emmert et al., 2020).*

**Author's response**:
We estimated our atomic oxygen uncertainties in the lower thermosphere by means of comparisons of our atomic oxygen input (WACCM) with other datasets (MSIS). We will try to better specify this in the paragraph.

Regarding the MSIS version, there was a mistake in the text. We actually used NRLMSIS 2.0 atomic oxygen (and not NRLMSISE-00) to estimate the uncertainties (see answer to the specific comment above).

Besides correcting the model version, we will re-order the sentence:

"Above 95 km, we have estimated the uncertainty in the atomic oxygen used in our retrievals, coming from WACCM, from comparisons with NRLMSIS 2.0 atomic oxygen (Emmert et al., 2020).

**Reviewer's suggestion**:
*Line 355-356. How is does the quenching rate used compare to what is used by SABER? Please state this quenching rate here and state both if different.*

**Author's response**:
They are the same. This is specified in Line 450 in Section 6: Comparison with SABER.

**Reviewer's suggestion**:
*Figure 9 caption. If the "M61" title is V8 and "M21" is V5 then please say so in the caption.*

**Author's response**:
We will do so.

**Reviewer's suggestion**:
*Lines 383-387. Given the importance of tidal variability, it would be important here to (re)state the equatorial crossing local times of the Envisat orbit.*

**Author's response**:
We will restate this in the revised version.

**Reviewer's suggestion**:
*Lines 431-432. Do the authors mean +/- 1000 km and +/- 2 hours? Please be explicit. Also, if "2 hours" means 2 hours in Universal Time rather than Local Time, the authors should say that as well. Given the importance of tides in the MLT, if "2 hours" means +/-2 hours UT then a statement in the text or the captions about the local times used for the MIPAS data and those used for the SABER data would also be useful.*

**Author's response**:
We will write "+/- 1000 km and +/- 2 hours UT'

**Reviewer's suggestion**:
*Line 450. Here it would also be helpful to the reader to also state how CO2 densities and O densities, as well as their diurnal variations, are specified in the MIPAS and SABER datasets. A few sentences with references would be very useful in diagnosing the comparisons between the two instruments. This is particularly important for altitudes above 100 km and also particularly important if the values used are different.*

**Author's response**:
In the revised version, we will specify the input used in SABER 2.0 retrievals, that is, $CO_2$ from WACCM3 climatologies, and O abundances derived from SABER ozone measurements below 95 km and taken from NRLMSISE-00 data above that altitude (Mlynczak et al., 2022 and references therein).

**Reviewer's suggestion**:
*Figures 12-15. In all of these figures, the reader cannot see the extent of the combined systematic errors at the top. Please either crop the figures at the top or expand the x-axes so the reader can see how big these are.*

**Author's response**:
We will expand the x-axes in all plots in Figures 12-15.

**Reviewer's suggestion**:
*Figures 12-15. To what does the "Nc" refer? If these are the number of coincident profiles, then please say so in the caption. Also, does that mean that the coincidences are pairs such that there are exactly Nc SABER profiles and Nc MIPAS profiles? If there are different numbers of*

*profiles that fall within the coincidence criteria then please state that explicitly. If the numbers of profiles are different for each instrument then please give both.*

**Author's response**:
"Nc" refers to the number of co-located individual MIPAS and SABER pairs. This will be specified in the caption of the figure in the revised version.

---

## Author Response (AR1)

**Referee #1**

*Reply to referee #1 for the comment on the manuscript* **amt-2023-119:**
**Version 8 IMK/IAA MIPAS temperatures from 12–15 µm spectra: Middle and Upper Atmosphere modes**

We thank the reviewer for his/her valuable comments and suggestions. We have addressed all of them. In this response, we go through the raised issues point by point and outline the changes we have made. Additionally, we have corrected several typos, and have re-worded or re-structured some sentences for clarity.

*Review of "Version 8 IMK/IAA MIPAS temperatures from 12-15 micron spectra: Middle and Upper Atmosphere modes" by Garcia-Comas et al.*

*General Comments:*

**Reviewer's summary:**
*This manuscript describes a new MIPAS temperature dataset from 18-115 km altitude, which the authors call version 8.03. In addition to improved calibrated spectra, the authors detail a variety of improvements to the retrieval and show several comparisons against previous versions of the data as well as comparisons against existing SABER temperature observations. The paper provides a comprehensive description of the uncertainties and overall, the figures are instructive and clear. Although the authors spend considerable effort comparing their results to SABER temperatures, comparisons with recently re-processed MIPAS temperatures using nitric oxide emission are quite limited and it is not really clear why. The manuscript is also lacking detail in some places, particularly in explaining the results in the upper altitude region of their dataset. The reviewer recommends publication provided that the authors address the concerns enumerated below.*

*Specific Comments:*

**Reviewer's suggestion:**
*Line 97 and elsewhere. The authors spend considerable effort comparing to existing SABER temperature observations. Could the authors please comment on and show how their results compare to MIPAS thermospheric temperatures derived from nitric oxide spectra at 5.3 microns (e.g. Funke et al., Atmos. Meas. Tech., 16, 2167-2196, 2023)? There is mention of day-night differences on lines 384-387 and latitude variations on lines 394-396. However, a direct comparison between the two MIPAS temperature datasets would be particularly useful for the reader. If there is a reason for this limited comparison or if this has been done in other publications then please say so and give a reference. Thank you.*

**Author's response:**
We think this is a very good suggestion.
We propose to rename Section 5 with "Consistency with NOM mode 12-15-micron and UA mode 5.3-micron temperatures" and to include the following paragraph at the end of Section 5:
"Thermospheric temperatures can also be derived from nitric oxide spectra in a different spectral region. Version 8 temperature retrievals from NO emissions at 5.3 microns, obtained by MIPAS in the UA mode, are detailed in \cite{Funke2023}. Below 110\,km, the a priori temperature profiles used to derive those 5.3-micron temperatures consist of the version 8

12—15-micron UA temperatures  presented here. Above 120 km, they use corrected NRLMSIS version 2.0 temperature profiles as a priori (Emmert et al., 2021). A merging of the two is performed at 110—120 km. Moreover, the 5.3-micron retrieved temperatures are strongly regularized towards our 12--15-micron UA temperatures below 110 km, as described in Funke et al. (2023).

We have compared the 5.3 μm with our 12–15 μm version 8 UA temperatures at 105–115 km (not shown), which is the altitude range where they concurrently provide significant information. Below 110 km, the 5.3 μm temperatures are essentially identical to our 12–15 μm temperatures for all atmospheric conditions, as expected due to the very strong regularization applied in the retrievals of the former. An exception to this occurs in the summer at latitudes poleward of 30∘, where the differences are within 5 K at 110 km. At 115 km, our 12–15 μm temperatures are generally 5–10 K higher than the 5.3 μm temperatures. The difference reaches 20–30 K for summer at latitudes poleward of 30∘ (note that the 5.3 um temperatures were already higher than NRLMSIS 2.0 temperatures in this altitude range (Funke et al., 2023)). Importantly, the differences do not generally depend on solar illumination. Nevertheless, the differences between the 5.3 μm and the 12–15 μm temperature datasets are within our estimated systematic errors for all atmospheric scenarios.”

**Reviewer's suggestion**:
*Lines 165-179. Could the authors show how the model results used for atomic oxygen compare to the recent MSIS 2.0 for atomic oxygen (Emmert et al., Earth and Space Science, 7, e2020EA001321. https://doi.org/10.1029/2020EA001321, 2020)? More importantly, how do any differences in MSIS 2.0 atomic oxygen affect the temperature retrieval?*

**Author's response**:
We thank the reviewer for this comment, that let us find a mistake in the text (Line 293). Section 3 of the manuscript describes the temperature error sources, including those coming from the uncertainties in the atomic oxygen abundances used in our retrievals. The estimation of the atomic oxygen uncertainty above 95 km has actually been done by means of comparisons with NRLMSIS 2.0 data, and not with NRMLMSISE00 data, as it was written in the text (lines 289-295). Therefore, Fig. 4 actually shows the average uncertainty based on comparisons with MSIS 2.0 atomic oxygen above 95km merged to the estimated MIPAS V5 atomic oxygen error below 95km (note that these are daytime and nighttime averages of the distinct abundances that we took for five latitude boxes and for the four seasons). Moreover, the effect of the differences between the atomic oxygen used and that of MSIS 2.0 on the retrieved temperatures is therefore included implicitly in the O-uncertainty component of our temperature non-LTE error estimation, which is described in Section 3.2 (see blue lines in Fig. 8). We would like to emphasize that, firstly, the effect of atomic oxygen uncertainty on retrieved temperature is only important above 95 km (see Fig. 8) and, secondly, the estimated MIPAS V5 atomic oxygen uncertainty is generally larger than that based on comparisons with MSIS 2.0 below 95 km.

We have corrected the MSIS model version number in the text and included the corresponding reference (Emmert et al., 2021) in the revised version.

**Reviewer's suggestion**:
*Line 230 and Figure 2. Are these kernels available to the reader? They would be useful for comparing with models or other datasets, but it is not clear from the data availability statement at the end of the paper that they are available. Note also that the reviewer could not connect using the link to the supplement at the end of the paper.*

**Author's response**:

Our data server does not regularly provide the individual averaging kernels for each temperature profile, due to the excessive volume of these data. Nevertheless, they can be provided upon request. In the data availability statement of the revised version, we now therefore state:" Due to their data volume, averaging kernels can be made available upon request."
The link to the supplement has been updated.

*Technical Corrections:*

**Reviewer's suggestion**:
*There are many acronyms throughout this abstract that are not spelled out. Please correct these when first mentioned. Thank you.*

**Author's response**:
We think we found all acronyms and spelled them out when first mentiones in the revised manuscript. We note here that, for an easier reading, we replaced "V8" and "V5" with "version 8" and "version 5" everywhere.

**Reviewer's suggestion**:
*Lines 25-26. Please give a number here (+/- X K) and indicate the altitudes at which this applies. SABER faces similar non-LTE challenges in that temperature retrieval. From Figures 12-15 the agreement gets worse and the combined systematic uncertainties get much larger above 90 km so the authors need to be more explicit about this in the abstract.*

**Author's response**:
We have replaced the last sentence in the abstract by the following sentences:
"The comparison of this temperature dataset with co-located Sounding of the Atmosphere using Broadband Emission Radiometry (SABER) temperature measurements shows an excellent agreement, with differences typically within 1.5 K below 90 km, 1–3 K at 90–95 km, 1–5 K at 95–100 km, 1–8 K at 100–105 km and 1–10 K above. The agreement with SABER improves with respect to previous MIPAS IMK/IAA data versions."

**Reviewer's suggestion**:
*Line 30. Please give a reference for these v5 spectra.*

**Author's response**:
We have included a reference in the revised version.

**Reviewer's suggestion**:
*Line 80. A brief paragraph describing the Envisat mission, as well as the orbital inclination and the equator crossing local times would be very helpful to the uninitiated reader here in order to give valuable context to the dataset.*

**Author's response**:
We have included the following sentences at the beginning of the second paragraph in Section 1:
"The ENVISAT satellite was placed into a polar Sun-synchronous orbit, with an inclination of 81.5º and an altitude of approximately 800 km. The orbital period was about 101 minutes, enabling the satellite to complete a global Earth coverage in 14.3 daily orbits. The Equator crossing local times were approximately 10:00 and 22:00."

**Reviewer's suggestion**:

*Line 149. What do the authors mean by "bias-corrected"? Is this a bias in the model and if so, why is this done? Given the importance of atomic oxygen in the temperature retrievals, one or two sentences explaining this would be helpful.*

**Author's response**:

We are not sure if the Line number indicated by the Reviewer is correct because that line describes the a priori temperature. Since the reviewer explicitly mentions atomic oxygen in this question, we think he refers to Line 167, where the bias-correction related to atomic oxygen is discussed.

The description of the "bias-correction" performed to the WACCM4 Ox to derive the atomic oxygen used in the temperature retrievals is actually written in the three paragraphs in Lines 168-179. To emphasize these all, describe the bias-correction, we have combined the three paragraphs into one.

In order to provide a justification for this correction, we have also included the following sentence:

"This approach is made to preserve the transient variability as provided by WACCM4 while retaining the climatological atomic oxygen as derived from MIPAS measurements in the previous version".

**Reviewer's suggestion**:

*Figure 2 caption. To what does 30 degrees refer?*

**Author's response**:

The words "latitude of" before the number were missing.

**Reviewer's suggestion**:

*Line 270. The reviewer does not understand how $CO_2$ uncertainties can be calculated from the WACCM results. Why can this uncertainty not be estimated by ACE and/or SABER data alone? Please explain in the text.*

**Author's response**:

The uncertainties are estimated from WACCM $CO_2$ (at MIPAS geolocations) by means of comparisons with ACE and SABER observations because that is the WACCM $CO_2$ abundance that we use as input for these temperature retrievals, as written in Line 163. Nevertheless, for clarity, we have added "at MIPAS geolocations" after WACCM $CO_2$ in this sentence.

**Reviewer's suggestion**:

*Lines 289-295. Again, the reviewer does not understand how an atomic oxygen uncertainty can be calculated from the WACCM results. Please explain in the text. Furthermore, it is well known that the NRLMSIS00 atomic oxygen values above 90 km are inaccurate (e.g. Sheese et al., J. Geophys. Res., 116, D01303, doi:10.1029/2010JD014640). The reviewer requests that the authors either crop their Fig. 4 at 97 km or use the newer more accurate values from MSIS 2.0 (Emmert et al., 2020).*

**Author's response**:

We estimated our atomic oxygen uncertainties in the lower thermosphere by means of comparisons of our atomic oxygen input (WACCM) with other datasets (MSIS). We tried to specify this better in this paragraph in the new version.

Regarding the MSIS version, there was a mistake in the text. We actually used NRLMSIS 2.0 atomic oxygen (and not NRLMSISE-00) to estimate the uncertainties (see answer to the specific comment above).

Besides correcting the model version, we have re-ordered the sentence:
"Above 95 km, we have estimated the uncertainty in the atomic oxygen used in our retrievals, coming from WACCM, from comparisons with NRLMSIS 2.0 atomic oxygen (Emmert et al., 2020).

**Reviewer's suggestion**:
*Line 355-356. How is does the quenching rate used compare to what is used by SABER? Please state this quenching rate here and state both if different.*

**Author's response**:
They are the same. This was specified in Line 450 in Section 6: Comparison with SABER.

**Reviewer's suggestion**:
*Figure 9 caption. If the "M61" title is V8 and "M21" is V5 then please say so in the caption.*

**Author's response**:
We changed the title of the plots to V8-V5.

**Reviewer's suggestion**:
*Lines 383-387. Given the importance of tidal variability, it would be important here to (re)state the equatorial crossing local times of the Envisat orbit.*

**Author's response**:
We have re-stated this in the revised version.

**Reviewer's suggestion**:
*Lines 431-432. Do the authors mean +/- 1000 km and +/- 2 hours? Please be explicit. Also, if "2 hours" means 2 hours in Universal Time rather than Local Time, the authors should say that as well. Given the importance of tides in the MLT, if "2 hours" means +/-2 hours UT then a statement in the text or the captions about the local times used for the MIPAS data and those used for the SABER data would also be useful.*

**Author's response**:
We have written "+/- 1000 km and +/- 2 hours UT (Universal Time)'

**Reviewer's suggestion**:
*Line 450. Here it would also be helpful to the reader to also state how CO2 densities and O densities, as well as their diurnal variations, are specified in the MIPAS and SABER datasets. A few sentences with references would be very useful in diagnosing the comparisons between the two instruments. This is particularly important for altitudes above 100 km and also particularly important if the values used are different.*

**Author's response**:
In the revised version, we have specified the input used in SABER 2.0 retrievals, that is, CO2 from WACCM3 climatologies, and O abundances derived from SABER ozone measurements below 95 km and taken from NRLMSISE-00 data above that altitude (Mlynczak et al., 2022 and references therein).

**Reviewer's suggestion**:
*Figures 12-15. In all of these figures, the reader cannot see the extent of the combined systematic errors at the top. Please either crop the figures at the top or expand the x-axes so the reader can see how big these are.*

**Author's response**:
We have expanded the x-axes in all plots in Figures 12-15. Note that this has required a different x-axes for polar summer plots.

**Reviewer's suggestion**:
*Figures 12-15. To what does the "Nc" refer? If these are the number of coincident profiles, then please say so in the caption. Also, does that mean that the coincidences are pairs such that there are exactly Nc SABER profiles and Nc MIPAS profiles? If there are different numbers of profiles that fall within the coincidence criteria then please state that explicitly. If the numbers of profiles are different for each instrument then please give both.*

**Author's response**:
"Nc" refers to the number of co-located individual MIPAS and SABER pairs. This is now specified in the caption of the figure in the revised version.

*Reply to referee #2 for the review of the manuscript*:
**Version 8 IMK/IAA MIPAS temperatures from 12–15 µm spectra: Middle and Upper Atmosphere modes (amt-2023-119)**

We thank the reviewer for his/her valuable comments and suggestions. We have addressed all of them. In this response, we go through the raised issues point by point and outline the changes we made. Additionally, we have corrected several typos, and have re-worded or re-structured some sentences for clarity.

**Reviewer's summary:**
*The paper reports the new retrieval of temperature in the mesosphere using MIPAS latest realeased level 1b spectra. I am not a native English speaker, therefore I just made few language corrections. It deserves to be published after the following comments have been addressed.*

**Reviewer's suggestion**:
*Line 29 : Add some reference to the ESA site or to the official document*

**Author's response**:
We have added a reference to the corresponding ESA file:
ESA: Product Quality Readme File for MIPAS Level 1b IPF 8.03 products – issue 1.1, ESA-EOPG-
   EBA-TN-1, available at:
   https://earth.esa.int/eogateway/documents/20142/37627/Read_Me_File_MIP_NL__1PY_ES
   A-EOPG-EBA-TN-1-issue1.1.pdf (last access: September 2023), 2019.

**Reviewer's suggestion**:
 *Line 34: This part is officially known as Optimised Resolution, please add this name when you first mention the part of the measurements*

**Author's response**:
We have added to the text: 'also called optimized resolution by ESA'.

**Reviewer's suggestion**:
*Line 39: substitute 'a few days in a row' with 'few (or write the number if it always the same) consecutive days'*

**Author's response**:
We have replaced it by 'few consecutive days' in the new version.

**Reviewer's suggestion**:
*Line 70: I assume each dataset refers to a single observation mode, I suggest making it clearer in the text here.*

**Author's response**:
That is correct. We have added 'respectively' after 'dataset'.

**Reviewer's suggestion**:

*Line 114-117: please add some reference to documents or papers that describe this.*

**Author's response**:
We have included the corresponding references.

**Reviewer's suggestion**:
*Line 124-125: there is a new release of HITRAN (2020) why did you not use it?*

**Author's response**:
Our processing of the MIPAS v8 data began in 2019, prior to the release of HITRAN2020. In that processing, not only temperature but other numerous atmospheric species were retrieved, for which this version 8 of temperatures were used. For consistency, the HITRAN2016 version has been maintained in all retrievals.

**Reviewer's suggestion**:
*Line 137: But you do not retrieve continuum above 58 km, so why you find problems?*

**Author's response**:
The reviewer is right to ask this question. In fact, the altitude was not correct: the heights where the offset correction cannot be distinguished are also below 60 km (in particular, in the regions where radiative transfer is linear). This sentence has been changed in the revised version, replacing the value of 60 km by ~30 km.

**Reviewer's suggestion**:
*Line 144: substitute 'km in just' with 'km just'*

**Author's response**:
Done.

**Reviewer's suggestion**:
*Line 154-155: The sentence should be changed in 'In V5R_t_m21, the temperature ….etc.'*

**Author's response**:
Done.

**Reviewer's suggestion**:
*Line 192: Substitute 'in each iteration' with 'at each iteration'*

**Author's response**:
Done.

**Reviewer's suggestion**:
*Line 193: Substitute 'change slightly' with 'have been slightly changed'*

**Author's response**:
Done.

**Reviewer's suggestion**:
*Line 200: Substitute 'is' with 'are'*

**Author's response**:
Done. Thank you.

**Reviewer's suggestion**:

*Line 208-210: Hard to see what you say in the figure due to the adopted color scale. Also checking the AK it looks that very small information is contained in the measurements above 100 km, so this comment doesn't hold very well*

**Author's response**:
We have changed the color scale and the contour lines. We think it is now easier to see. This is particularly true for the plots for solstice, where the isolines are inclined down towards the polar summer.
We also think there is considerable information up to 110 km in MA and NLC (averaging kernel diagonal ~0.18) and up to 115 km in UA (averaging kernel diagonal ~0.26), as the averaging kernels in Fig. 2 show. As an action to this comment and another one below, we have added the following sentence:
"Our recommendation for the data users is to discard data points at altitudes where the corresponding averaging kernel diagonal element is less than 0.03, because they lack significant measurement information."

**Reviewer's suggestion**:

*Lines 213-215: Again very hard to see in the figure*

**Author's response**:
This refers to the yellowish blob around 85 km at low latitudes during equinox. We now mention this explicitly in the text to make it easier to find.

**Reviewer's suggestion**:

*Lines 219-222: Could you please explain better this point, I could not follow it! What 'elevated stratopause events' are?*

**Author's response**:
Elevated stratopauses were described in detail in Manney et al. (2005). The term 'Elevated stratopause' refer to extreme events in which the stratopause reforms at uncommonly high altitudes (in the middle-upper mesosphere), occasionally occuring after sudden stratospheric warmings ( see e.g., Manney et al. (2005), Siskind et al., 2007; Chandran et al., 2011, 2013; McLandress et al., 2013; Limpasuvan et al., 2016). We have included the reference to Manney et al. (2005) in the revised version:
Manney, G. L., Krüger, K., Sabutis, J. L., Sena, S. A., and Pawson, S.: The remarkable 2003–2004 winter and other recent warm winters in the Arctic stratosphere since the late 1990s, J. Geophys. Res., 110, D04107, https://doi.org/10.1029/2004JD005367, 2005.

**Reviewer's suggestion**:

*Line 224: 'wrinkled' is not very scientific, use oscilating or something more appropriate.*

**Author's response**:
We chose "wrinkled" because we wanted to emphasize that it is an irregular structure. We would use "oscillating" to denote a regular structure. With this in mind and if the reviewer agrees, we prefer to stick to "wrinkled" in this context.

**Reviewer's suggestion**:

*Line 230: what is the threshold you use of the AK peak to decide if there is information?*

**Author's response**:

We should have written the threshold in the text, as the reviewer suggests. We have included the following sentence in the revised version:

"Our recommendation for the data users is to discard data points at altitudes where the corresponding averaging kernel diagonal element is less than 0.03, because they lack significant measurement information."

**Reviewer's suggestion**:

*Lines 235-239: Could you explain why the vertical resolution oscillates so much?*

**Author's response**:

This is a natural phenomenon characteristic of the retrieval when using more retrieved points (state vector) and measurements. and there is nothing wrong with it: at a tangent altitude (where the instrument provides measurements), the vertical resolution is better but the error by measurement noise is larger. Between two tangent altitudes, where the results depend on information of two measurements (above and below), noise is smaller but the resolution is worse. If averaging kernels were
evaluated only on the grid given by the tangent altitudes, this effect would not be visible.
We have now included a sentence in the revised manuscript clarifying this behavior:
"The oscillating behavior depicted arises because the vertical resolution is better at retrieval altitudes closer to the tangent altitudes (see Fig. 4), despite the larger error introduced by measurement noise."

**Reviewer's suggestion**:

*Line 245: you say 34 scenarios: shouldn't them be 40? 5 latitude bands X 4 seasons X 2 (day/night)*

**Author's response**:

Please, note that we do not distinguish between seasons for tropical latitudes.

**Reviewer's suggestion**:

*Line 249: You say 'in the following', is it in the table or in the text?*

**Author's response**:

The expression was certainly not clear. In the revised version, we have replaced: "In the following, we provide 1-sigma uncertainties." with "All reported uncertainties are one sigma throughout."

**Reviewer's suggestion**:

*Line 253: 'propagation of measurement' -> 'propagation of the measurement'*

**Author's response**:

We agree. We have changed the text accordingly.

**Reviewer's suggestion**:

*Line 255: The noise values change very much from the region around 700 cm-1 to the region at 900 cm-1, I would rather use the range than the average value here*

**Author's response**:

We now provide the noise range in the revised version. We would like to note that we do not use the entire band up to 900 cm-1.

**Reviewer's suggestion**:

*Lines 256-259: The calibration is performed as part of your analysis or as part of the level1 process?*

**Author's response**:
In the revised version, we now specify that: "(…) there is still a remaining random uncertainty due to the wavelength dependence of the deep space measurements used for the level-1b radiance offset calibration."

**Reviewer's suggestion**:
*Lines 260-261: do you analyse unapodised spectra? Otherwise the ILS shape should be dominated by the apodization used*

**Author's response**:
We use apodized spectra, and this apodization is considered in the error estimation. It still does make an effect, particularly at higher altitudes.

**Reviewer's suggestion**:
*Table 3: even if you explain it in the caption the word chief does not suggest what you have introduced in the column, could you use a different and more appropriate word?*

**Author's response**:
We have substituted 'chief' by 'Char.' (from Characteristic) in the column title.

**Reviewer's suggestion**:
*Lines 275-278: HITRAN usually reports spectroscopic uncertainties, why haven't you used them?*

**Author's response**:
Unfortunately, HITRAN spectroscopic uncertainties are not completely helpful because it is not reported whether they are 1-sigma, 2-sigma or 3-sigma. Contacting the HITRAN lead author did not help either. Thus, we decided to contact an expert in the field to provide these uncertainties (Manfred Birk, February 2020), as it was stated in the text.

**Reviewer's suggestion**:
*Lines 279-283: the whole paragraph is rather confusing, could you explain it better? An educated guess is not a quantity, could you explicit the guess you made?*

**Author's response**:
We tried to be clearer in the revised version. We now specify the uncertainty used for the calculations. We have also re-written the paragraph:
"For gases derived in preceding MIPAS V5 retrievals, we have used the noise covariance information obtained from those retrievals. For interfering gases not previously retrieved from MIPAS measurements, we relied on our initial guess database (Kiefer et al., 2002, and updates thereof). In the latter case, accurate uncertainty information for these abundances is often unavailable and we have assumed a 100\% uncertainty."

**Reviewer's suggestion**:
*Lines 296-297: Some of the random errors are systematic, but of a random nature….*

**Author's response**:
We are not certain if we have interpreted this comment correctly. In the context of data analysis and, consequently, for the benefit of data users, we believe it is advantageous to

provide systematic and random errors separately. For instance, systematic errors result in biases that need not be considered when analyzing variations relative to mean values. This is discussed in Clarmann et al. (2020), as referenced in the text, and is beyond the scope of this manuscript.

**Reviewer's suggestion**:
*Lines 305-306: Do you mean that you did not know the covariance matrix associated to those errors?*

**Author's response**:
It depends on the error source. For instance, there was no covariance information available for CO2. However, for most of the other error sources that we treated using the perturbation method, the covariance treatment is unnecessary since the uncertainties pertain to single, independent parameters. In other words, there would be little gained by using the covariance formalism because we are only dealing with the variances of the single, independent parameters.

In order to be more precise, we have add "due to unavailability of covariance information" after "not possible" in the revised version.

**Reviewer's suggestion**:
*Lines 307-309: The whole sentence is not very clear, could you describe this better?*

**Author's response**:
We propose to re-write the sentence as follows:
"There were error sources for which no specific values of their random uncertainty could be prescribed (for instance, CO2 abundance and spectroscopy, non-LTE and ILS). Since temperature errors triggered by a systematic parameter uncertainty also exhibit a random component due to atmospheric variability, we also conducted statistical analyses on the responses to perturbations across the ensemble of temperature profiles, each ensemble representing a distinct atmospheric scenario. We took the dispersion within these responses as our estimate of the associated temperature error random component for those cases."

**Reviewer's suggestion**:
*Line 313: what do you mean with 'chiefly'?*

**Author's response**:
We mean 'not completely, but as a most important part', a synonym of 'mainly' or 'primarily'. Although we think both "Chiefly random errors" and "chiefly systematic errors" are grammatically and semantically correct, we have changed the word order in this sentence to write "Errors chiefly random" and "errors chiefly systematic", in the hope that the expression can be better understood.

**Reviewer's suggestion**:
*Line 315 'for 34' -> 'for the 34'*

**Author's response**:
Done.

**Reviewer's suggestion**:
*Line 430: Which NLTE model does Saber use? If it is the same as in this paper it should be mentioned, as in this altitude range it is the main contributor to the retrieved temperatures.*

**Author's response**:
The references where the SABER non-LTE model is described are provided in the manuscript (Remsberg et al., 2008; Garcia-Comas et al. 2008). The SABER non-LTE algorithm is not the same as the MIPAS non-LTE model. However, some non-LTE parameters are set equal. That is the case for the quenching rates of $CO_2(v_2)$ by O, N2 and O2, but, for instance, not the case for atomic oxygen abundance or the rate of $CO_2-CO_2$ $v_2$-quanta exchange. Consequently, for our SABER and MIPAS comparisons, we removed the contribution from the CO2-O, CO2-N2 and CO2-O2 quenching rate uncertainties from the combined errors and the discussion in the text takes this fact into account. This was specifically stated in lines 447-450.

**Reviewer's suggestion**:
*Line 481 onward: I did understand that the comparison were done on the smoothed profiles only, so please clarify this better in the text!*

**Author's response**:
We propose to change the wording:
"Beyond this, we have also compared collocated MIPAS and SABER time series. For this purpose, MIPAS averaging kernels and a priori information were applied to the SABER profiles."

**Reviewer's suggestion**:
*In the references at Line 699 ' Fera, S.D.' -> ' Della Fera, S.'*

**Author's response**:
Thank you. We have changed this name in the reference.